# Convex and Bilevel Optimization for Neuro-Symbolic Inference and Learning

## Abstract

We address a key challenge for neuro-symbolic (NeSy) systems by leveraging convex and bilevel optimization techniques to develop a general gradient-based framework for end-to-end neural and symbolic parameter learning. The applicability of our framework is demonstrated with NeuPSL, a state-of-the-art NeSy architecture. To achieve this, we propose a smooth primal and dual formulation of NeuPSL inference and show learning gradients are functions of the optimal dual variables. Additionally, we develop a dual block coordinate descent algorithm for the new formulation that naturally exploits warm-starts. This leads to over $100\times$ learning runtime improvements over the current best NeuPSL inference method. Finally, we provide extensive empirical evaluations across 8 datasets covering a range of tasks and demonstrate our learning framework achieves up to a $16\%$ point prediction performance improvement over alternative learning methods.

## 1 Introduction

The goal of neuro-symbolic (NeSy) AI is a seamless integration of neural models for processing low-level data with symbolic frameworks to reason over high-level symbolic structures (d'Avila Garcez et al., 2002; 2009; 2019). This paper addresses an important research challenge in NeSy with the introduction of a principled and general NeSy learning framework. Further, we propose a novel inference algorithm and establish theoretical properties for a state-of-the-art NeSy system that are crucial for learning.

Our proposed learning framework builds upon NeSy energy-based models (NeSy-EBMs) (Pryor et al., 2023), a general class of NeSy systems that encompasses a variety of existing NeSy methods, including DeepProblog (Manhaeve et al., 2018; 2021), SATNet (Wang et al., 2019), logic tensor networks (Badreddine et al., 2022), and NeuPSL (Pryor et al., 2023). NeSy-EBMs use neural network outputs to parameterize an energy function and formulate an inference problem that may be non-smooth and constrained. Thus, predictions are not guaranteed to be a function of the inputs and parameters with an explicit form or to be differentiable, and traditional deep learning techniques are not directly applicable. We therefore equivalently formulate NeSy-EBM learning as a bilevel problem and, to support smooth first-order gradient-based optimization, propose a smoothing strategy that is novel to NeSy learning. Specifically, we replace the constrained NeSy energy function with its Moreau envelope. The augmented Lagrangian method for equality-constrained minimization is then applied with the new formulation.

We demonstrate the effectiveness of our proposed learning framework with NeuPSL. To ensure differentiability and provide principled forms of gradients for learning, we present a new formulation and regularization of NeuPSL inference as a quadratic program. Moreover, we introduce a dual block coordinate descent (dual BCD) inference algorithm for the quadratic program. The dual BCD algorithm is the first NeuPSL inference method that produces optimal dual variables for producing both optimal primal variables and gradients for learning. Additionally, empirical results demonstrate that dual BCD is able to effectively leverage warm starts, thus improving learning runtime.

Our key contributions are: (1) An improved formulation of the NeSy-EBM learning problem that establishes a foundation for applying smooth first-order gradient-based optimization techniques; (2) A reformulation of NeuPSL inference that is used to prove continuity properties and obtain explicit forms of gradients for learning; (3) A dual BCD algorithm for NeuPSL inference that naturally

produces statistics necessary for computing gradients for learning and that fully leverages warm-starts to improve learning runtime; (4) Two parallelization strategies for dual BCD inference; and (5) A thorough empirical evaluation demonstrating prediction performance improvements on 8 different datasets and a learning runtime speedup of up to $100\times$.

## 2 RELATED WORK

NeSy AI is an active area of research that incorporates symbolic (commonly logical and arithmetic) reasoning with neural networks (Bader & Hitzler, 2005; d'Avila Garcez et al., 2009; Besold et al., 2017; De Raedt et al., 2020; Lamb et al., 2020; Giunchiglia et al., 2022). We will show that learning for a general class of NeSy systems is naturally formulated as bilevel optimization (Bracken & McGill, 1973; Colson et al., 2007; F. Bard, 2013). In other words, the NeSy learning objective is a function of predictions obtained by solving a lower-level inference problem that is symbolic reasoning. In this work, we focus on a general setting where the lower-level problem is an expressive and complex program capable of representing cyclic dependencies and ensuring the satisfaction of constraints during both learning and inference (Wang et al., 2019; Badreddine et al., 2022; Dasarth et al., 2023; Pryor et al., 2023; Cornelio et al., 2023). One prominent and tangential subgroup of such NeSy systems we would like to acknowledge enforces constraints on the structure of the symbolic model, and hence the lower-level problem, to ensure the final prediction has an explicit gradient with respect to the parameters (Xu et al., 2018; Manhaeve et al., 2021; Ahmed et al., 2022). In the deep learning community, bilevel optimization also arises in hyperparameter optimization and meta-learning (Pedregosa, 2016; Franceschi et al., 2018), generative adversarial networks (Goodfellow et al., 2014), and reinforcement learning (Sutton & Barto, 2018).

Researchers typically take one of three approaches to bilevel optimization: (1) *Implicit differentiation* methods compute or approximate the Hessian matrix at the lower-level problem solution to derive an analytic expression for the gradient of the upper-level objective called a hypergradient (Do et al., 2007; Pedregosa, 2016; Ghadimi & Wang, 2018; Rajeswaran et al., 2019; Giovannelli et al., 2022; Khanduri et al., 2023). (2) *Automatic differentiation* methods unroll inference into a differentiable computational graph (Stoyanov et al., 2011; Domke, 2012; Belanger et al., 2017; Ji et al., 2021). (3) *Value-Function approaches* reformulate the bilevel problem as a single-level constrained program using the optimal value of the lower-level objective (the *value-function*) to develop principled first-order gradient-based algorithms that do not require the calculation of Hessian matrices for the lower-level problem (V. Outrata, 1990; Liu et al., 2021; Sow et al., 2022; Liu et al., 2022; 2023; Kwon et al., 2023). Note that standard algorithms for all three approaches to bilevel optimization suggest solving the lower-level problem to derive the gradients used for optimizing the bilevel program. Principled techniques for using approximate lower-level solutions to make progress on the bilevel program is an open research direction (Pedregosa, 2016; Liu et al., 2021). Further, the lower-level problem for NeSy learning (inference) is commonly constrained. Implicit differentiation methods have been developed for bilevel optimization with lower-level constraints (Giovannelli et al., 2022; Khanduri et al., 2023). We introduce a value-function approach.

## 3 NESY ENERGY-BASED MODELS

In this work, we use *NeSy energy-based models (NeSy-EBMs)* (Pryor et al., 2023) to develop a generally applicable NeSy learning framework. Here, we provide background on NeSy-EBMs and introduce a classification of losses that motivates the need for general learning algorithms.

NeSy-EBMs are a family of EBMs (LeCun et al., 2006) that use neural model predictions to define potential functions with symbolic interpretations. NeSy-EBM energy functions are parameterized by a set of neural and symbolic weights from the domains $\mathcal{W}_{nn}$ and $\mathcal{W}_{sy}$, respectively, and quantify the compatibility of a target variable from a domain $\mathcal{Y}$ and neural and symbolic inputs from the domains $\mathcal{X}_{nn}$ and $\mathcal{X}_{sy}$: $E : \mathcal{Y} \times \mathcal{X}_{sy} \times \mathcal{X}_{nn} \times \mathcal{W}_{sy} \times \mathcal{W}_{nn} \to \mathbb{R}$. NeSy-EBM inference requires first computing the output of the neural networks, *neural inference*, and then minimizing the energy function over the targets, *symbolic inference*:

$$\arg\min_{\mathbf{y} \in \mathcal{Y}} E(\mathbf{y}, \mathbf{x}_{sy}, \mathbf{x}_{nn}, \mathbf{w}_{sy}, \mathbf{w}_{nn}). \tag{1}$$

NeSy-EBM learning is finding weights to create an energy function that associates lower energies to target values near their truth in a set of training data. The training data consists of $P$ samples that

are tuples of symbolic variables and neural network inputs: $\{S_1 = (\mathbf{y}_1, \mathbf{x}_{1,sy}, \mathbf{x}_{1,nn}), \cdots, S_P = (\mathbf{y}_P, \mathbf{x}_{P,sy}, \mathbf{x}_{P,nn})\}$. Moreover, targets $\mathbf{y}_i$ from a training sample $S_i$ are partitioned into *labeled variables*, $\mathbf{t}_i$, for which there is a corresponding truth value, and *latent variables*, $\mathbf{z}_i$. Without loss of generality, we write $\mathbf{y}_i = (\mathbf{t}_i, \mathbf{z}_i)$. NeSy-EBM learning losses are defined using the *latent minimizer*, $\mathbf{z}_i^* \in \arg\min_{\mathbf{z} \in \mathcal{Z}} E((\mathbf{t}_i, \mathbf{z}), \mathbf{x}_{i,sy}, \mathbf{x}_{i,nn}, \mathbf{w}_{sy}, \mathbf{w}_{nn})$, the *full minimizer*, $\mathbf{y}_i^* \in \arg\min_{\mathbf{y} \in \mathcal{Y}} E(\mathbf{y}, \mathbf{x}_{i,sy}, \mathbf{x}_{i,nn}, \mathbf{w}_{sy}, \mathbf{w}_{nn})$, and the *latent* and *full optimal value-functions*:

$$V_{\mathbf{z}_i^*}(\mathbf{w}_{sy}, \mathbf{w}_{nn}) := E((\mathbf{t}_i, \mathbf{z}_i^*), \mathbf{x}_{i,sy}, \mathbf{x}_{i,nn}, \mathbf{w}_{sy}, \mathbf{w}_{nn}), \tag{2}$$

$$V_{\mathbf{y}_i^*}(\mathbf{w}_{sy}, \mathbf{w}_{nn}) := E(\mathbf{y}_i^*, \mathbf{x}_{i,sy}, \mathbf{x}_{i,nn}, \mathbf{w}_{sy}, \mathbf{w}_{nn}). \tag{3}$$

Note the optimal values-functions are functions of the parameters, inputs, and symbolic variables; however, to simplify notation, we only write the parameters as arguments.

*Value-based* learning losses depend on the model weights strictly via the optimal value-functions. Two common value-based losses for NeSy-EBMs are the latent optimal value-function (*energy loss*), and the difference between the latent and full optimal value-functions (*structured perceptron loss*) (LeCun et al., 1998; Collins, 2002):

$$L_{Energy}(E(\cdot, \cdot, \cdot, \mathbf{w}_{sy}, \mathbf{w}_{nn}), S_i) := V_{\mathbf{z}_i^*}(\mathbf{w}_{sy}, \mathbf{w}_{nn}), \tag{4}$$

$$L_{SP}(E(\cdot, \cdot, \cdot, \mathbf{w}_{sy}, \mathbf{w}_{nn}), S_i) := V_{\mathbf{z}_i^*}(\mathbf{w}_{sy}, \mathbf{w}_{nn}) - V_{\mathbf{y}_i^*}(\mathbf{w}_{sy}, \mathbf{w}_{nn}). \tag{5}$$

A principled first-order gradient-based method for optimizing a value-based objective only requires differentiability of the value-functions. However, performance metrics are not always aligned with value-based losses. Moreover, they are known to have degenerate solutions, e.g., weights minimizing the loss but producing a collapsed energy function (LeCun et al., 2006; Pryor et al., 2023).

Alternatively, *minimizer-based* learning losses assume the minimizer of the energy function is unique. With this assumption, energy minimization is a vector-valued function from the weight space $\mathcal{W}_{sy} \times \mathcal{W}_{nn}$ to the target space $\mathcal{Y}$, $\mathbf{y}_i^*(\mathbf{w}_{sy}, \mathbf{w}_{nn}) : \mathcal{W}_{sy} \times \mathcal{W}_{nn} \to \mathcal{Y}$. Then, minimizer-based losses are compositions of a differentiable supervised loss $d : \mathcal{Y} \times \mathcal{Y} \to \mathbb{R}$, and the minimizer:

$$L_d(E(\cdot, \cdot, \cdot, \mathbf{w}_{sy}, \mathbf{w}_{nn}), S_i) := d(\mathbf{y}_i^*(\mathbf{w}_{sy}, \mathbf{w}_{nn}), \mathbf{t}_i). \tag{6}$$

Minimizer-based losses are general and allow learning with objectives aligned with evaluation metrics. However, a direct application of a first-order gradient based method for minimizer-based learning requires the Jacobian at the minimizer. NeSy-EBM predictions are not necessarily differentiable. Even if they are differentiable, the computation of the Jacobian is often too expensive to be practical.

## 4 A BILEVEL NESY LEARNING FRAMEWORK

In this section, we introduce a general framework for the bilevel NeSy learning problem:

$$\arg\min_{\substack{(\mathbf{w}_{sy}, \mathbf{w}_{nn}) \in \mathcal{W}_{sy} \times \mathcal{W}_{nn} \\ (\mathbf{y}_1, \cdots, \mathbf{y}_P) \in \mathcal{Y} \times \cdots \times \mathcal{Y}}} \sum_{i=1}^{P} \left( d(\mathbf{y}_i, \mathbf{t}_i) + L_{Val}(E(\cdot, \cdot, \cdot, \mathbf{w}_{sy}, \mathbf{w}_{nn}), S_i) \right) + \mathcal{R}(\mathbf{w}_{sy}, \mathbf{w}_{nn}) \tag{7}$$

$$\text{s.t.} \quad \mathbf{y}_i \in \arg\min_{\mathbf{y} \in \mathcal{Y}} E(\mathbf{y}, \mathbf{x}_{i,sy}, \mathbf{x}_{i,nn}, \mathbf{w}_{sy}, \mathbf{w}_{nn}), \quad \forall i \in \{1, \cdots, P\}$$

where $d$ and $L_{Val}$ are a minimizer and value-based loss, respectively, and $\mathcal{R} : \mathcal{W}_{sy} \times \mathcal{W}_{nn} \to \mathbb{R}$ is a regularizer. We make the following (standard) lower-level singleton assumption.

**Assumption 4.1.** $E$ is minimized over $\mathbf{y} \in \mathcal{Y}$ at a single point for every $(\mathbf{w}_{sy}, \mathbf{w}_{nn}) \in \mathcal{W}_{sy} \times \mathcal{W}_{nn}$.

Under Assumption 4.1, and regardless of the continuity and curvature properties of the upper and lower level objectives, (7) is equivalent to the following:

$$\arg\min_{\substack{(\mathbf{w}_{sy}, \mathbf{w}_{nn}) \in \mathcal{W}_{sy} \times \mathcal{W}_{nn} \\ (\mathbf{y}_1, \cdots, \mathbf{y}_P) \in \mathcal{Y} \times \cdots \times \mathcal{Y}}} \sum_{i=1}^{P} \left( d(\mathbf{y}_i, \mathbf{t}_i) + L_{Val}(E(\cdot, \cdot, \cdot, \mathbf{w}_{sy}, \mathbf{w}_{nn}), S_i) \right) + \mathcal{R}(\mathbf{w}_{sy}, \mathbf{w}_{nn}) \tag{8}$$

$$\text{s.t.} \quad E(\mathbf{y}_i, \mathbf{x}_{i,sy}, \mathbf{x}_{i,nn}, \mathbf{w}_{sy}, \mathbf{w}_{nn}) - V_{\mathbf{y}_i^*}(\mathbf{w}_{sy}, \mathbf{w}_{nn}) \le 0, \quad \forall i \in \{1, \cdots, P\}.$$

The formulation in (8) is referred to as a *value-function* approach in bilevel optimization literature (V. Outrata, 1990; Liu et al., 2021; 2022; Sow et al., 2022; Kwon et al., 2023). Value-function approaches view the bilevel program as a single-level constrained optimization problem by leveraging the value-function as a tight lower bound on the lower-level objective. However, the inequality

constraints in (8) do not satisfy any of the standard *constraint qualifications* that ensure the feasible set near the optimal point is similar to its linearized approximation (Nocedal & Wright, 2006). This raises a challenge for providing theoretical convergence guarantees for constrained optimization techniques. Following a recent line of value-function approaches to bilevel programming (Liu et al., 2021; Sow et al., 2022; Liu et al., 2023), we overcome this challenge by allowing at most an $\iota > 0$ violation in each constraint in (8). With this relaxation, strictly feasible points exist and, for instance, the linear independence constraint qualification (LICQ) can hold.

Another challenge that arises from (8) is that the energy function of NeSy-EBMs is typically non-differentiable with respect to the targets and even infinite-valued to implicitly represent constraints. As a result, penalty or augmented Lagrangian functions derived from (8) are intractable. Therefore, we substitute each instance of the energy function evaluated at the training sample $i$ and parameterized by $(\mathbf{w}_{sy}, \mathbf{w}_{nn})$ in the constraints of (8) with the following function:

$$M_i(\mathbf{y}; \mathbf{w}_{sy}, \mathbf{w}_{nn}, \rho) := \inf_{\hat{\mathbf{y}} \in \mathcal{Y}} \left( E(\hat{\mathbf{y}}, \mathbf{x}_{i,sy}, \mathbf{x}_{i,nn}, \mathbf{w}_{sy}, \mathbf{w}_{nn}) + \frac{1}{2\rho} \|\hat{\mathbf{y}} - \mathbf{y}\|_2^2 \right), \qquad (9)$$

where $\rho$ is a positive scalar. For convex $E$, (9) is the Moreau envelope of the energy function Rockafellar (1970); Parikh & Boyd (2013). In general, even for non-convex energy functions, the smoothing in (9) preserves global minimizers and minimum values, i.e., $\mathbf{y}_i^*(\mathbf{w}_{sy}, \mathbf{w}_{nn}) = \arg\min_{\mathbf{y}} M_i(\mathbf{y}; \mathbf{w}_{sy}, \mathbf{w}_{nn}, \rho)$ and $V_{\mathbf{y}_i^*}(\mathbf{w}_{sy}, \mathbf{w}_{nn}) = \min_{\mathbf{y}} M_i(\mathbf{y}; \mathbf{w}_{sy}, \mathbf{w}_{nn}, \rho)$. Moreover, under Assumption 4.1 each $M_i$ is finite for all $\mathbf{y} \in \mathcal{Y}$ even if the energy function is not. When the energy function is a lower semi-continuous convex function, its Moreau envelope is convex, finite, and continuously differentiable, and its gradient with respect to $\mathbf{y}$ is:

$$\nabla_{\mathbf{y}} M_i(\mathbf{y}; \mathbf{w}_{sy}, \mathbf{w}_{nn}, \rho) = \frac{1}{\rho} \left( \mathbf{y} - \arg\min_{\hat{\mathbf{y}} \in \mathcal{Y}} \left( \rho E(\hat{\mathbf{y}}, \mathbf{x}_{i,sy}, \mathbf{x}_{i,nn}, \mathbf{w}_{sy}, \mathbf{w}_{nn}) + \frac{1}{2} \|\hat{\mathbf{y}} - \mathbf{y}\|_2^2 \right) \right) \qquad (10)$$

Convexity is a sufficient but not necessary condition to ensure each $M_i$ is differentiable with respect to the target variables. See Bonnans & Shapiro (2000) for results regarding the sensitivity of optimal value-functions to perturbations.

We propose the following relaxed and smoothed value-function approach to finding an approximate solution of (7):

$$\arg\min_{\substack{(\mathbf{w}_{sy}, \mathbf{w}_{nn}) \in \mathcal{W}_{sy} \times \mathcal{W}_{nn} \\ (\mathbf{y}_1, \cdots, \mathbf{y}_P) \in \mathcal{Y} \times \cdots \times \mathcal{Y}}} \sum_{i=1}^{P} \left( d(\mathbf{y}_i, \mathbf{t}_i) + L_{Val}(E(\cdot, \cdot, \mathbf{w}_{sy}, \mathbf{w}_{nn}), S_i) \right) + \mathcal{R}(\mathbf{w}_{sy}, \mathbf{w}_{nn}) \qquad (11)$$

$$\text{s.t.} \qquad M_i(\mathbf{y}_i; \mathbf{w}_{sy}, \mathbf{w}_{nn}, \rho) - V_{\mathbf{y}_i^*}(\mathbf{w}_{sy}, \mathbf{w}_{nn}) \leq \iota, \quad \forall i \in \{1, \cdots, P\},$$

The formulation (11) is the core of our proposed NeSy-EBM learning framework outlined in Algorithm 1. The algorithm proceeds by approximately solving instances of (11) in a sequence defined by a decreasing $\iota$. This is a graduated approach to solving (8) with instances of (11) that are increasingly tighter approximations. Each instance of (11) is optimized using only first-order gradients of the energy and value-functions with the bound-constrained augmented Lagrangian algorithm, Al-

---

**Algorithm 1** NeSy-EBM Learning Framework

**Require:** Constraint Tolerance: $\sigma^*$, Movement Tolerance: $\omega^*$, Moreau Param.: $\rho$, Starting points: $(\mathbf{w}_{sy}^{(0)}, \mathbf{w}_{nn}^{(0)}) \in \mathcal{W}_{sy} \times \mathcal{W}_{nn}$

1: $\mathbf{y}_i^{(0)} \leftarrow (\mathbf{t}_i, \mathbf{z}_i^*), \forall i = 1, \cdots, P$;
2: $\iota^{(0)} \leftarrow \max_{i \in \{1, \cdots, P\}} M_i(\mathbf{y}_i^{(0)}; \mathbf{w}_{sy}^{(0)}, \mathbf{w}_{nn}^{(0)}, \rho) - V_{\mathbf{y}_i^*}(\mathbf{w}_{sy}^{(0)}, \mathbf{w}_{nn}^{(0)})$;

3: **for** $t = 0, 1, 2, \cdots$ **do**
4:      Find $\mathbf{w}_{sy}^{(t+1)}, \mathbf{w}_{nn}^{(t+1)}, \mathbf{y}_1^{(t+1)}, \cdots, \mathbf{y}_P^{(t+1)}$ minimizing (11) with $\iota^{(t)}$.
5:      **if** Stopping criterion satisified **then**
6:          Stop with: $\mathbf{w}_{sy}^{(t+1)}, \mathbf{w}_{nn}^{(t+1)}, \mathbf{y}_1^{(t+1)}, \cdots, \mathbf{y}_P^{(t+1)}$;
7:      $\iota^{(t+1)} \leftarrow \frac{1}{2} \cdot \iota^{(t)}$;

---

gorithm 17.4 from Nocedal & Wright (2006). Specifically, the algorithm finds approximate minimizers of the problem's augmented Lagrangian for a fixed setting of the penalty parameters using gradient descent. To simplify notation, let the equality constraints in (11) be denoted by:

$$c_i(\mathbf{y}_i, \mathbf{w}_{sy}, \mathbf{w}_{nn}; \iota) := M_i(\mathbf{y}_i; \mathbf{w}_{sy}, \mathbf{w}_{nn}, \rho) - V_{\mathbf{y}_i^*}(\mathbf{w}_{sy}, \mathbf{w}_{nn}) - \iota,$$

for each constraint indexed $i \in \{1, \cdots, P\}$. Moreover, let $c(\mathbf{y}_1, \cdots, \mathbf{y}_P, \mathbf{w}_{sy}, \mathbf{w}_{nn}; \iota) := [c_i(\mathbf{y}_i, \mathbf{w}_{sy}, \mathbf{w}_{nn}; \iota)]_{i=1}^P$. The augmented Lagrangian function corresponding to (11) introduces a quadratic penalty parameter $\mu$ and $P$ linear penalty parameters $\lambda := [\lambda_i]_{i=1}^P$, as follows:

$$\mathcal{L}_A(\mathbf{w}_{sy}, \mathbf{w}_{nn}, \mathbf{y}_1, \cdots, \mathbf{y}_p, \mathbf{s}; \lambda, \mu, \iota) := \sum_{i=1}^{P} \left( d(\mathbf{y}_i, \mathbf{t}_i) + L_{Val}(E(\cdot, \cdot, \mathbf{w}_{sy}, \mathbf{w}_{nn}), S_i) \right)$$

$$+ \frac{\mu}{2} \sum_{i=1}^{P} \left( c_i(\mathbf{y}_i, \mathbf{w}_{sy}, \mathbf{w}_{nn}; \iota) + s_i \right)^2 + \sum_{i=1}^{P} \lambda_i \left( c_i(\mathbf{y}_i, \mathbf{w}_{sy}, \mathbf{w}_{nn}; \iota) + s_i \right) + \mathcal{R}(\mathbf{w}_{sy}, \mathbf{w}_{nn}). \qquad (12)$$

where we introduced $P$ slack variables, $\mathbf{s} = [s_i]_{i=1}^P$, for each inequality constraint. We make the following assumption to ensure the augmented Lagrangian function is differentiable:

**Assumption 4.2.** Every $V_{y_i^*}$, $V_{z_i^*}$, and $M_i$ is differentiable with respect to the weights.

We employ the bound-constrained augmented Lagrangian algorithm to solve (11) (see Appendix B for details). This method provides a principled algorithm for updating the penalty parameters and ensures fundamental convergence properties of our learning framework. Notably, we have that limit points of the iterate sequence are stationary points of $\|c(\mathbf{y}_1, \cdots, \mathbf{y}_P, \mathbf{w}_{sy}, \mathbf{w}_{nn}) + \mathbf{s}\|^2$ when the problem has no feasible points. When the problem is feasible and LICQ holds at the limits, they are KKT points of (11) (Theorem 17.2 in Nocedal & Wright (2006)). Convergence rates and stronger guarantees are likely possible from analyzing the structure of the energy function for specific NeSy-EBMs and is a direction for future work.

The value for $\iota$ is halved every time an approximate solution to the Lagrangian subproblem is reached. We suggest starting points for each $\mathbf{y}_i^{(0)}$ to be the latent inference minimizer and $\iota^{(0)}$ to be the maximum difference in the value-function and the smooth energy function over all $\mathbf{y}_i^{(0)}$. The outer loop of the NeSy-EBM learning framework may be stopped by either watching the progress of a training or validation evaluation metric, or by specifying a final value for $\iota$.

## 5 NEUPSL AND DEEP HINGE-LOSS MARKOV RANDOM FIELDS

We demonstrate the applicability of our learning framework with Neural Probabilistic Soft Logic (NeuPSL), a general class of NeSy-EBMs designed for scalable joint reasoning (Pryor et al., 2023). In NeuPSL, relations and attributes are represented by *atoms*, and dependencies between atoms are encoded with first-order logical clauses and linear arithmetic inequalities referred to as *rules*. Atom values can be target variables, observations, or outputs from a neural network. The rules and atoms are translated into potentials measuring rule satisfaction and are aggregated to define a member of a tractable class of graphical models: *deep hinge-loss Markov random fields* (deep HL-MRF).

**Definition 5.1.** Let $\mathbf{g} = [g_i]_{i=1}^{n_g}$ be functions with corresponding weights $\mathbf{w}_{nn} = [\mathbf{w}_{nn,i}]_{i=1}^{n_g}$ and inputs $\mathbf{x}_{nn}$ such that $g_i : (\mathbf{w}_{nn,i}, \mathbf{x}_{nn}) \mapsto [0,1]$. Let $\mathbf{y} \in [0,1]^{n_y}$ and $\mathbf{x}_{sy} \in [0,1]^{n_x}$. A **deep hinge-loss potential** is a function of the form:

$$\phi(\mathbf{y}, \mathbf{x}_{sy}, \mathbf{g}(\mathbf{x}_{nn}, \mathbf{w}_{nn})) := (\max\{\mathbf{a}_{\phi,\mathbf{y}}^T \mathbf{y} + \mathbf{a}_{\phi,\mathbf{x}_{sy}}^T \mathbf{x}_{sy} + \mathbf{a}_{\phi,\mathbf{g}}^T \mathbf{g}(\mathbf{x}_{nn}, \mathbf{w}_{nn}) + b_\phi, 0\})^p,$$

where $\mathbf{a}_{\phi,\mathbf{y}} \in \mathbb{R}^{n_y}$, $\mathbf{a}_{\phi,\mathbf{x}} \in \mathbb{R}^{n_x}$, and $\mathbf{a}_{\phi,\mathbf{g}} \in \mathbb{R}^{n_g}$ are variable coefficient vectors, $b_\phi \in \mathbb{R}$ is a vector of constants, and $p \in \{1, 2\}$. Let $\mathcal{T} = [\tau_i]_{i=1}^r$ denote an ordered partition of a set of $m$ deep hinge-loss potentials. Further, define $\mathbf{\Phi}(\mathbf{y}, \mathbf{x}_{sy}, \mathbf{g}(\mathbf{x}_{nn}, \mathbf{w}_{nn})) := [\sum_{k \in \tau_i} \phi_k(\mathbf{y}, \mathbf{x}_{sy}, \mathbf{g}(\mathbf{x}_{nn}, \mathbf{w}_{nn}))]_{i=1}^r$. Let $\mathbf{w}_{sy}$ be a vector of $r$ non-negative symbolic weights corresponding to the partition $\mathcal{T}$. Then, a **deep hinge-loss energy function** is:

$$E(\mathbf{y}, \mathbf{x}_{sy}, \mathbf{x}_{nn}, \mathbf{w}_{sy}, \mathbf{w}_{nn}) := \mathbf{w}_{sy}^T \mathbf{\Phi}(\mathbf{y}, \mathbf{x}_{sy}, \mathbf{g}(\mathbf{x}_{nn}, \mathbf{w}_{nn})). \tag{13}$$

Let $\mathbf{a}_{c_k,\mathbf{y}} \in \mathbb{R}^{n_y}$, $\mathbf{a}_{c_k,\mathbf{x}} \in \mathbb{R}^{n_x}$, $\mathbf{a}_{c_k,\mathbf{g}} \in \mathbb{R}^{n_g}$, and $b_{c_k} \in \mathbb{R}$ for each $k \in 1, \ldots, q$ and $q \geq 0$ be vectors defining linear inequality constraints and a feasible set:

$$\mathbf{\Omega}(\mathbf{x}_{sy}, \mathbf{g}) := \left\{ \mathbf{y} \in [0,1]^{n_y} \mid \mathbf{a}_{c_k,\mathbf{y}}^T \mathbf{y} + \mathbf{a}_{c_k,\mathbf{x}}^T \mathbf{x}_{sy} + \mathbf{a}_{c_k,\mathbf{g}}^T \mathbf{g} + b_{c_k} \leq 0, \forall k = 1, \ldots, q \right\}.$$

Then a **deep hinge-loss Markov random field** defines the conditional probability density:

$$P(\mathbf{y}|\mathbf{x}_{sy}, \mathbf{x}_{nn}) := \begin{cases} \frac{\exp(-E(\mathbf{y}, \mathbf{x}_{sy}, \mathbf{x}_{nn}, \mathbf{w}_{sy}, \mathbf{w}_{nn}))}{\int_{\mathbf{y} \in \mathbf{\Omega}(\cdot)} \exp(-E(\mathbf{y}, \mathbf{x}_{sy}, \mathbf{x}_{nn}, \mathbf{w}_{sy}, \mathbf{w}_{nn})) d\mathbf{y}} & \mathbf{y} \in \mathbf{\Omega}(\mathbf{x}_{sy}, \mathbf{g}(\mathbf{x}_{nn}, \mathbf{w}_{nn})) \\ 0 & o.w. \end{cases} \tag{14}$$

NeuPSL inference is finding the MAP state of the conditional distribution defined by a deep HL-MRF, i.e., finding the minimizer of the energy function over the feasible set.

$$\min_{\mathbf{y} \in \mathbb{R}^{n_\mathbf{y}}} \mathbf{w}_{sy}^T \mathbf{\Phi}(\mathbf{y}, \mathbf{x}_{sy}, \mathbf{g}(\mathbf{x}_{nn}, \mathbf{w}_{nn})) \quad \text{s.t. } \mathbf{y} \in \mathbf{\Omega}(\mathbf{x}_{sy}, \mathbf{g}(\mathbf{x}_{nn}, \mathbf{w}_{nn})). \tag{15}$$

As each of the potentials are convex, (15) is a non-smooth convex linearly constrained program.

## 5.1 A SMOOTH FORMULATION OF INFERENCE

In this section, we introduce a primal and dual formulation of NeuPSL inference as a linearly constrained convex quadratic program (LCQP). (See Appendix C.1 for details.) In summary, $m$ slack variables with lower bounds and $2 \cdot n_{\mathbf{y}} + m$ linear constraints are defined to represent the target variable bounds and deep hinge-loss potentials. All $2 \cdot n_{\mathbf{y}} + m$ variable bounds, $m$ potentials, and $q \geq 0$ constraints are collected into a $(2 \cdot n_{\mathbf{y}} + q + 2 \cdot m) \times (n_{\mathbf{y}} + m)$ dimensional matrix $\mathbf{A}$ and a vector of $(2 \cdot n_{\mathbf{y}} + q + 2 \cdot m)$ elements that is an affine function of the neural predictions and symbolic inputs $\mathbf{b}(\mathbf{x}_{sy}, \mathbf{g}(\mathbf{x}_{nn}, \mathbf{w}_{nn}))$. Moreover, the slack variables and a $(n_{\mathbf{y}} + m) \times (n_{\mathbf{y}} + m)$ positive semi-definite diagonal matrix, $\mathbf{D}(\mathbf{w}_{sy})$, and a $(n_{\mathbf{y}} + m)$ dimensional vector, $\mathbf{c}(\mathbf{w}_{sy})$, are created using the symbolic weights to define a quadratic objective. Further, we gather the original target variables and the slack variables into a vector $\nu \in \mathbb{R}^{n_{\mathbf{y}} + m}$. Altogether, the regularized convex LCQP reformulation of NeuPSL inference is:

$$V(\mathbf{w}_{sy}, \mathbf{b}(\mathbf{x}_{sy}, \mathbf{g}(\mathbf{x}_{nn}, \mathbf{w}_{nn}))) := \tag{16}$$
$$\min_{\nu \in \mathbb{R}^{n_{\mathbf{y}}+m}} \nu^T (\mathbf{D}(\mathbf{w}_{sy}) + \epsilon \mathbf{I})\nu + \mathbf{c}(\mathbf{w}_{sy})^T \nu \quad \text{s.t.} \ \mathbf{A}\nu + \mathbf{b}(\mathbf{x}_{sy}, \mathbf{g}(\mathbf{x}_{nn}, \mathbf{w}_{nn})) \leq 0,$$

where $\epsilon \geq 0$ is a scalar regularization parameter added to the diagonal of $\mathbf{D}$ to ensure strong convexity (needed in the next subsection). The effect of the added regularization is empirically studied in Appendix E.3. The function $V(\mathbf{w}_{sy}, \mathbf{b}(\mathbf{x}_{sy}, \mathbf{g}(\mathbf{x}_{nn}, \mathbf{w}_{nn})))$ in (16) is the optimal value-function of the LCQP formulation of NeuPSL inference referred to in the previous section.

By Slater's constraint qualification, we have strong duality when there is a feasible solution to (16). In this case, an optimal solution to the dual problem yields an optimal solution to the primal problem. The Lagrange dual problem of (16) is:

$$\min_{\mu \in \mathbb{R}^{2 \cdot n_{\mathbf{y}}+m+q}_{\geq 0}} h(\mu; \mathbf{w}_{sy}, \mathbf{b}(\mathbf{x}_{sy}, \mathbf{g}(\mathbf{x}_{nn}, \mathbf{w}_{nn}))) \tag{17}$$
$$:= \frac{1}{4}\mu^T \mathbf{A}(\mathbf{D}(\mathbf{w}_{sy}) + \epsilon \mathbf{I})^{-1}\mathbf{A}^T \mu + \frac{1}{2}(\mathbf{A}(\mathbf{D}(\mathbf{w}_{sy}) + \epsilon \mathbf{I})^{-1}\mathbf{c}(\mathbf{w}_{sy}) - 2\mathbf{b}(\mathbf{x}_{sy}, \mathbf{g}(\mathbf{x}_{nn}, \mathbf{w}_{nn})))^T \mu,$$

where $\mu$ is the vector of dual variables and $h(\mu; \mathbf{w}_{sy}, \mathbf{b}(\mathbf{w}_{nn}))$ is the LCQP dual objective function. As $(\mathbf{D}(\mathbf{w}_{sy}) + \epsilon \mathbf{I})$ is diagonal, it is easy to invert, and thus it is practical to work in the dual space and map dual to primal variables. The dual-to-primal variable mapping is:

$$\nu \leftarrow -\frac{1}{2}(\mathbf{D}(\mathbf{w}_{sy}) + \epsilon \mathbf{I})^{-1}(\mathbf{A}^T \mu + \mathbf{c}(\mathbf{w}_{sy})). \tag{18}$$

On the other hand, the primal-to-dual mapping is more computationally expensive and requires calculating a pseudo-inverse of the constraint matrix $\mathbf{A}$.

## 5.2 CONTINUITY OF INFERENCE

We use the LCQP formulation in (16) to establish continuity and curvature properties of the NeuPSL energy minimizer and the optimal value-function provided in the following theorem. The proof is provided in Appendix C.2.

**Theorem 5.2.** *Suppose for any setting of $\mathbf{w}_{nn} \in \mathbb{R}^{n_g}$ there is a feasible solution to NeuPSL inference (16). Further, suppose $\epsilon > 0$, $\mathbf{w}_{sy} \in \mathbb{R}^r_+$, and $\mathbf{w}_{nn} \in \mathbb{R}^{n_g}$. Then:*

- *The minimizer of (16), $\mathbf{y}^*(\mathbf{w}_{sy}, \mathbf{w}_{nn})$, is a $O(1/\epsilon)$ Lipschitz continuous function of $\mathbf{w}_{sy}$.*
- $V(\mathbf{w}_{sy}, \mathbf{b}(\mathbf{x}_{sy}, \mathbf{g}(\mathbf{x}_{nn}, \mathbf{w}_{nn})))$, *is concave over $\mathbf{w}_{sy}$ and convex over $\mathbf{b}(\mathbf{x}_{sy}, \mathbf{g}(\mathbf{x}_{nn}, \mathbf{w}_{nn}))$.*
- $V(\mathbf{w}_{sy}, \mathbf{b}(\mathbf{x}_{sy}, \mathbf{g}(\mathbf{x}_{nn}, \mathbf{w}_{nn})))$ *is differentiable with respect to $\mathbf{w}_{sy}$. Moreover,*
$$\nabla_{\mathbf{w}_{sy}} V(\mathbf{w}_{sy}, \mathbf{b}(\mathbf{x}_{sy}, \mathbf{g}(\mathbf{x}_{nn}, \mathbf{w}_{nn}))) = \mathbf{\Phi}(\mathbf{y}^*(\mathbf{w}_{sy}, \mathbf{w}_{nn}), \mathbf{x}_{sy}, \mathbf{g}(\mathbf{x}_{nn}, \mathbf{w}_{nn})).$$
  *Furthermore, $\nabla_{\mathbf{w}_{sy}} V(\mathbf{w}_{sy}, \mathbf{b}(\mathbf{x}_{sy}, \mathbf{g}(\mathbf{x}_{nn}, \mathbf{w}_{nn})))$ is Lipschitz continuous over $\mathbf{w}_{sy}$.*
- *If there is a feasible point $\nu$ strictly satisfying the $i'th$ inequality constraint of (16), i.e., $\mathbf{A}[i]\nu + \mathbf{b}(\mathbf{x}_{sy}, \mathbf{g}(\mathbf{x}_{nn}, \mathbf{w}_{nn}))[i] < 0$, then $V(\mathbf{w}_{sy}, \mathbf{b}(\mathbf{x}_{sy}, \mathbf{g}(\mathbf{x}_{nn}, \mathbf{w}_{nn})))$ is subdifferentiable with respect to the $i'th$ constraint constant $\mathbf{b}(\mathbf{x}_{sy}, \mathbf{g}(\mathbf{x}_{nn}, \mathbf{w}_{nn}))[i]$. Moreover,*
$$\partial_{\mathbf{b}[i]} V(\mathbf{w}_{sy}, \mathbf{b}(\mathbf{x}_{sy}, \mathbf{g}(\mathbf{x}_{nn}, \mathbf{w}_{nn}))) = \{\mu^*[i] \mid \mu^* \in \arg\min_{\mu \in \mathbb{R}^{2 \cdot n_{\mathbf{y}}+m+q}_{\geq 0}} h(\mu; \mathbf{w}_{sy}, \mathbf{b}(\mathbf{x}_{sy}, \mathbf{g}(\mathbf{x}_{nn}, \mathbf{w}_{nn})))\}.$$

  *Furthermore, if $\mathbf{g}(\mathbf{x}_{nn}, \mathbf{w}_{nn})$ is a smooth function of $\mathbf{w}_{nn}$, then so is $\mathbf{b}(\mathbf{x}_{sy}, \mathbf{g}(\mathbf{x}_{nn}, \mathbf{w}_{nn}))$, and the set of regular subgradients of $V(\mathbf{w}_{sy}, \mathbf{b}(\mathbf{x}_{sy}, \mathbf{g}(\mathbf{x}_{nn}, \mathbf{w}_{nn})))$ is:*
$$\hat{\partial}_{\mathbf{w}_{nn}} V(\mathbf{w}_{sy}, \mathbf{b}(\mathbf{x}_{sy}, \mathbf{g}(\mathbf{x}_{nn}, \mathbf{w}_{nn}))) \tag{19}$$
$$\supset \nabla_{\mathbf{w}_{nn}} \mathbf{b}(\mathbf{x}_{sy}, \mathbf{g}(\mathbf{x}_{nn}, \mathbf{w}_{nn}))^T \partial_{\mathbf{b}} V(\mathbf{w}_{sy}, \mathbf{b}(\mathbf{x}_{sy}, \mathbf{g}(\mathbf{x}_{nn}, \mathbf{w}_{nn}))).$$

Theorem 5.2 provides a simple explicit form of the value-function gradient with respect to the symbolic weights and regular subgradient with respect to the neural weights. Moreover, this result is directly applicable to the Moreau envelope of the NeuPSL energy function used in Section 4 as it is a regularized value-function. Thus, Theorem 5.2 supports the principled application of Algorithm 1 for learning both the symbolic and neural weights of a NeuPSL model.

### 5.3 DUAL BLOCK COORDINATE DESCENT

The regular subgradients in Theorem 5.2 are functions of the optimal dual variables of the LCQP inference problem in (17). For this reason, we introduce a block coordinate descent (BCD) (Wright, 2015) algorithm for working directly with the dual LCQP formulation of inference. Details of the algorithm are provided in Appendix D. Our dual BCD algorithm is the first method specialized for the dual LCQP inference and is therefore also the first to produce optimal dual variables that directly yeild both optimal primal variables and principled gradients for learning, all without the need to compute a pseudo-inverse of the constraint matrix.

The dual BCD algorithm proceeds by successively minimizing the objective along the subgradient of a block of dual variables. For this reason, dual BCD guarantees descent at every iteration, partially explaining its effectiveness at leveraging warm-starts and improving learning runtimes. The algorithm is stopped when the primal-dual gap drops below a threshold $\delta > 0$. We suggest a practical choice of variable blocks with efficient methods for computing the objective subgradients and solving the steplength subproblems. Additionally, we develop an efficient method for identifying connected components of the factor graph defined by the deep HL-MRF yeilding a variable partition that the dual objective is additively separable over to parallelize the BCD updates. Moreover, inspired by lock-free parallelization strategies (Bertsekas & N. Tsitsiklis, 1989; Recht et al., 2011; Liu et al., 2015), we also propose a variant of the dual BCD inference algorithm that sacrifices the theoretical guaranteed descent property for significant runtime improvements. In Section 6, we show that the lock-free dual BCD algorithm consistently finds a solution satisfying the stopping criterion, and surprisingly, is still highly effective at leveraging warm starts.

## 6 EMPIRICAL EVALUATION

We evaluate the runtime and prediction performance of our proposed NeSy inference and parameter learning algorithms on the 8 datasets in Table 1[1]. The table includes the dataset's inference task, the associated prediction performance metric, and whether the corresponding NeuPSL model has deep neural network parameters. Unless noted

Table 1: Datasets used for empirical evaluations.

| Dataset | Deep | Task | Perf. Metric |
|---|---|---|---|
| CreateDebate Hasan & Ng (2013) | | Stance Class. | AUROC |
| 4Forums Walker et al. (2012) | | Stance Class. | AUROC |
| Epinions (Richardson et al., 2003) | | Link Pred. | AUROC |
| DDI (S. Wishart et al., 2006) | | Link Pred. | AUROC |
| Yelp (Yelp, 2023) | | Regression | MAE |
| Citeseer (Sen et al., 2008) | ✓ | Node Class. | Accuracy |
| Cora (Sen et al., 2008) | ✓ | Node Class. | Accuracy |
| MNIST-Add.(Manhaeve et al., 2018) | ✓ | Image Class. | Accuracy |

otherwise, all experiments are run on 5 splits and the average and standard deviation of times and performance metric values are reported. Details on the datasets, hardware specifications, hyperparameter searches, and model architectures are provided in Appendix E.

For learning experiments in Section 6.2 and Section 6.3, NeuPSL models with weights trained using value-based learning losses, e.g., energy and structured perceptron (SP), use mirror descent (Kivinen & Warmuth, 1997; Shalev-Shwartz, 2012) on the symbolic weights constrained to the unit simplex and Adam (P. Kingma & Lei Ba, 2017) for the neural weights. NeuPSL models with weights trained using minimizer-based losses, e.g., mean squared error (MSE) and binary cross entropy (BCE), use our proposed NeSy learning framework in Algorithm 1 with a scaled energy loss term added to the objective as in (7). Moreover, optimization of the augmented Lagrangian, line 4 of Algorithm 1, is performed using the bound constrained augmented Lagrangian algorithm (Appendix B) with mirror descent on the symbolic weights and Adam for the neural weights.

---

[1]All code and data is available at `https://github.com/convexbilevelnesylearning`.

## 6.1 INFERENCE RUNTIME

We begin by examining the runtime of symbolic inference. We evaluate the alternating direction method of multipliers (ADMM) Boyd et al. (2010), the current state-of-the-art inference algorithm for NeuPSL, and our proposed inference algorithms: connected component parallel dual BCD (CC D-BCD) and lock-free parallel dual BCD (LF D-BCD). We also evaluate the performance of Gurobi, a leading off-the-shelf optimizer, and subgradient descent (GD) in Appendix E.4. All inference algorithms have access to the same computing resources . We run a hyperparameter search,

Table 2: Time in seconds for inference using ADMM and our proposed CC D-BCD and LF D-BCD algorithms on each dataset.

|  | ADMM | CC D-BCD | LF D-BCD |
|---|---|---|---|
| CreateDebate | $9.98 \pm 1.13$ | $\mathbf{0.05 \pm 0.02}$ | $0.05 \pm 0.03$ |
| 4Forums | $15.17 \pm 0.74$ | $0.11 \pm 0.02$ | $\mathbf{0.05 \pm 0.01}$ |
| Epinions | $0.36 \pm 0.041$ | $1.84 \pm 0.4$ | $\mathbf{0.26 \pm 0.04}$ |
| Citeseer | $0.63 \pm 0.07$ | $1.36 \pm 0.24$ | $\mathbf{0.49 \pm 0.08}$ |
| Cora | $\mathbf{0.71 \pm 0.07}$ | $6.46 \pm 3.5$ | $0.79 \pm 0.19$ |
| DDI | $7.85 \pm 0.28$ | $31.47 \pm 0.17$ | $\mathbf{1.76 \pm 0.17}$ |
| Yelp | $\mathbf{6.37 \pm 1.19}$ | $48.44 \pm 3.82$ | $7.58 \pm 0.48$ |
| MNIST-Add1 | $11.45 \pm 1.32$ | $\mathbf{10.23 \pm 1.04}$ | $115 \pm 45$ |
| MNIST-Add2 | $285 \pm 66$ | $\mathbf{29.09 \pm 8.00}$ | $1,189 \pm 16$ |

detailed in Appendix E.4, for each algorithm, and the configuration yielding a prediction performance that is within a standard deviation of the best and completed with the lowest runtime is reported. All algorithms are stopped when the $L_\infty$ norm of the primal variable change between iterates is less than 0.001.

The total average inference runtime in seconds for each algorithm and model is provided in Table 2. Surprisingly, despite the potential for an inexact solution to the BCD steplength subproblem, LF D-BCD is faster than CC D-BCD in the first 7 datasets and demonstrates up to $6\times$ speedup over CC D-BCD in Yelp. However, in MNIST-Add datasets, CC D-BCD is up to $10\times$ faster than LF D-BCD as there is a high number of tightly connected components, one for each addition instance. This behavior highlights the complementary strengths of the two parallelization strategies. LF D-BCD should be applied to problems with larger factor graph representations that are connected while CC D-BCD is effective when there are many similarly sized connected components.

## 6.2 LEARNING RUNTIME

Next, we study how the algorithms applied to solve inference affect the learning runtime with the SP and MSE losses. Specifically, we examine the cumulative time required for ADMM and D-BCD inference to complete 500 weight updates on the first 7 datasets in Table 1 and 100 weight updates on MNIST-Add datasets. Hyperparameters used for SP and MSE learning are reported in Appendix E.5. For inference, we apply the

Table 3: Cumulative time in seconds for ADMM and D-BCD inference during learning with SP and MSE losses.

|  | SP | | MSE | |
|---|---|---|---|---|
|  | ADMM | D-BCD | ADMM | D-BCD |
| CreateDebate | $10.68 \pm 8.63$ | $\mathbf{0.34 \pm 0.36}$ | $49.00 \pm 31.23$ | $\mathbf{0.62 \pm 0.09}$ |
| 4Forums | $11.87 \pm 12.81$ | $\mathbf{0.65 \pm 0.05}$ | $67.09 \pm 13.79$ | $\mathbf{1.11 \pm 0.16}$ |
| Epinions | $12.54 \pm 0.37$ | $\mathbf{1.33 \pm 0.06}$ | $17.48 \pm 0.62$ | $\mathbf{2.27 \pm 0.98}$ |
| Citeseer | $167 \pm 37$ | $\mathbf{41.57 \pm 6.39}$ | $225 \pm 32$ | $\mathbf{70.01 \pm 5.86}$ |
| Cora | $183 \pm 26$ | $\mathbf{48.16 \pm 5.82}$ | $241 \pm 37$ | $\mathbf{79.62 \pm 13.77}$ |
| DDI | $4,554 \pm 13$ | $\mathbf{19.65 \pm 0.30}$ | $7,652 \pm 218$ | $\mathbf{52.78 \pm 4.23}$ |
| Yelp | $1,835 \pm 47$ | $\mathbf{114 \pm 4}$ | $2,250 \pm 100$ | $\mathbf{170 \pm 12}$ |
| MNIST-Add1 | $1,624 \pm 34$ | $\mathbf{232 \pm 44}$ | $2,942 \pm 109$ | $\mathbf{2,738 \pm 93}$ |
| MNIST-Add2 | TIME-OUT | $\mathbf{804 \pm 106}$ | TIME-OUT | $\mathbf{4,291 \pm 114}$ |

same hyperparameters used in the previous section and the fastest parallelization method for D-BCD.

Table 3 shows that the D-BCD algorithm consistently results in the lowest total inference runtime, validating it's ability to leverage warm starts to improve learning runtimes. Notably, on the DDI dataset, D-BCD achieves roughly a $100\times$ speedup over ADMM. Moreover, on MNIST-Add2, ADMM timed out with over 6 hours of inference time for SP and MSE learning, while D-BCD accumulated less than 0.5 and 1.2 hours of inference runtime on average for SP and MSE, respectively.

## 6.3 LEARNING PREDICTION PERFORMANCE

In our final experiment, we analyze the prediction performance of NeuPSL models trained with our NeSy-EBM learning framework. A hyperparameter search (detailed in Appendix E.6) is performed over learning steplengths, regularizations, and parameters for Algorithm 1.

**HL-MRF learning** We first evaluate the prediction performance on non-deep vari-

Table 4: Prediction performance of HL-MRF models trained on value and minimizer-based losses.

|  | Energy | SP | MSE | BCE |
|---|---|---|---|---|
| CreateDebate | $64.76 \pm 9.54$ | $64.68 \pm 11.05$ | $\mathbf{65.33 \pm 11.98}$ | $64.83 \pm 9.70$ |
| 4Forums | $62.96 \pm 6.11$ | $63.15 \pm 6.40$ | $64.22 \pm 6.41$ | $\mathbf{64.85 \pm 6.01}$ |
| Epinions | $78.96 \pm 2.29$ | $79.85 \pm 1.62$ | $\mathbf{81.18 \pm 2.21}$ | $80.89 \pm 2.32$ |
| Citeseer | $70.29 \pm 1.54$ | $70.92 \pm 1.33$ | $71.22 \pm 1.56$ | $\mathbf{71.94 \pm 1.17}$ |
| Cora | $54.30 \pm 1.74$ | $74.16 \pm 2.32$ | $81.05 \pm 1.41$ | $\mathbf{81.07 \pm 1.31}$ |
| DDI | $94.54 \pm 0.00$ | $94.61 \pm 0.00$ | $94.70 \pm 0.00$ | $\mathbf{95.08 \pm 0.00}$ |
| Yelp | $18.11 \pm 0.34$ | $18.57 \pm 0.66$ | $18.14 \pm 0.36$ | $\mathbf{17.93 \pm 0.50}$ |

ants of NeuPSL models for the first 7 datasets, i.e., only symbolic weights are learned. Table 4 shows that across all 7 datasets, NeuPSL models trained with Algorithm 1 obtain a better average prediction performance than those trained using a valued-based loss. On the Cora dataset, the NeuPSL model fit with the BCE loss achieves over a $6\%$ point improvement over SP, the higher-performing value-based loss.

**Deep HL-MRF learning** Next, we evaluate the prediction performance of deep NeuPSL models. Here, we study the standard low-data setting for Citeseer and Cora. Specifically, results are averaged over 10 randomly sampled splits using $5\%$ of the nodes for training, $5\%$ of the nodes for validation, and $1,000$ nodes for testing. We also report the prediction performance of the same strong baseline models used in Pryor et al. (2023) for this task: DeepStochLog (Winters et al., 2022), and a Graph Convolutional Network (GCN) (Kipf & Welling, 2017). Additionally, we investigate performance on MNIST-Addition, a widely used NeSy evaluation task first introduced by Manhaeve et al. (2018). In MNIST-Addition, models must determine the sum of two lists of MNIST images, for example, $(\lceil 3 \rceil + \lceil 5 \rceil = 8)$. The challenge stems from the lack of labels for the MNIST images; only the final sum of the equation is provided during training, 8 in this example. Implementation details for the neural and symbolic components of the NeuPSL models for both citation network and MNIST-Add experiments are provided in Appendix E.6.

Table 5: Accuracy of DeepStochlog, GCN, and NeuPSL on Citeseer and Cora.

| | | | | NeuPSL | | | |
|---|---|---|---|---|---|---|---|
| | **DeepStochlog** | **GCN** | **Energy** | **SP** | **MSE** | **BCE** |
| **Citeseer** | $62.68 \pm 3.84$ | $67.42 \pm 0.66$ | $69.63 \pm 1.33$ | $\mathbf{69.78 \pm 1.42}$ | $69.62 \pm 1.27$ | $69.64 \pm 1.33$ |
| **Cora** | $71.28 \pm 1.98$ | $80.32 \pm 1.11$ | $80.41 \pm 1.81$ | $78.59 \pm 4.93$ | $\mathbf{81.48 \pm 1.45}$ | $81.28 \pm 1.45$ |

Table 6: Accuracy of CNN, LTN, DeepProblog and NeuPSL on MNIST-Addition.

| | | | | | NeuPSL | |
|---|---|---|---|---|---|---|
| | **Additions** | **CNN** | **LTN** | **DeepProblog** | **Energy** | **BCE** |
| **MNIST-Add1** | 300 | $17.16 \pm 00.62$ | $69.23 \pm 15.68$ | $85.61 \pm 01.28$ | $87.96 \pm 01.58$ | $\mathbf{88.84 \pm 02.07}$ |
| | 3,000 | $78.99 \pm 01.14$ | $93.90 \pm 00.51$ | $92.59 \pm 01.40$ | $95.60 \pm 0.91$ | $\mathbf{95.70 \pm 0.84}$ |
| **MNIST-Add2** | 150 | $01.31 \pm 00.23$ | $02.02 \pm 00.97$ | $71.37 \pm 03.90$ | $59.20 \pm 32.79$ | $\mathbf{76.00 \pm 2.61}$ |
| | 1,500 | $01.69 \pm 00.27$ | $71.79 \pm 27.76$ | $87.44 \pm 02.15$ | $90.56 \pm 0.61$ | $\mathbf{93.04 \pm 2.26}$ |

Table 5 shows that fitting the neural network weights of a NeuPSL model with our NeSy-EBM learning framework is effective. NeuPSL models fit with the MSE and BCE losses consistently outperform both DeepStochlog and the GCN baseline. Moreover, Table 6 demonstrates NeuPSL models trained with Algorithm 1 and a BCE loss can achieve up to a $16\%$ point performance improvement over those trained with a value-based loss.

# 7 LIMITATIONS

Our learning framework is limited to NeSy-EBMs satisfying the two assumptions made in Section 4. While we advance the theory for NeuPSL to show it meets the assumptions, we do not know how to support NeSy-EBMs with non-differentiable value-functions. One approach is to substitute the inference program with a principled approximation. Lastly, although the idea to leverage inference algorithms such as BCD that effectively use warm-starts and improve learning runtimes is general, the inference algorithms were implemented for a NeSy system with an LCQP structure.

# 8 CONCLUSIONS AND FUTURE WORK

We introduced a general learning framework for NeSy-EBMs and demonstrated its applicability with NeuPSL. Additionally, we proposed a novel NeuPSL inference formulation and algorithm with practical and theoretical advantages. A promising direction for future work is to extend the learning framework to support approximate inference solutions for estimating the objective gradient to further improve learning runtimes. In addition, the empirical results presented in this work motivate generalizing and applying our learning framework to additional NeSy systems and tasks.

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

## A  APPENDIX

This appendix includes the following sections: extended bilevel NeSy Learning framework, extended NeuPSL and deep hinge-loss Markov random fields, extended dual block coordinate descent, and extended empirical evaluation.

Code for running the experiments along with all data, models, and hyperparameters are available at: `https://github.com/convexbilevelnesylearning/experimentscripts`. Code for the NeuPSL implementation of our proposed learning framework and inference algorithms is available at: `https://github.com/convexbilevelnesylearning/psl`.

## B  EXTENDED BILEVEL NESY LEARNING FRAMEWORK

In this section we provide the further details on our proposed NeSy learning framework. A complete version of Algorithm 1 is provided in Algorithm 2.

---

**Algorithm 2** Full NeSy-EBM Learning Framework

---

**Require:** Constraint Tolerance: $\sigma^*$, Movement Tolerance: $\omega^*$, Moreau Param.: $\rho$
  Starting points: $\mu^{(0)} > 1$, $\lambda_1^{(0)}, \cdots, \lambda_P^{(0)}$, $(\mathbf{w}_{sy}^{(0)}, \mathbf{w}_{nn}^{(0)}) \in \mathcal{W}_{sy} \times \mathcal{W}_{nn}$

1:  $\mathbf{y}_i^{(0)} \leftarrow (\mathbf{t}_i, \mathbf{z}_i^*)$, $\forall i = 1, \cdots, P$;
2:  $\iota^{(0)} \leftarrow \max_{i \in \{1, \cdots, P\}} M_i(\mathbf{y}_i^{(0)}; \mathbf{w}_{sy}^{(0)}, \mathbf{w}_{nn}^{(0)}, \rho) - V_{\mathbf{y}_i^*}(\mathbf{w}_{sy}^{(0)}, \mathbf{w}_{nn}^{(0)})$;
3:  **for** $t = 0, 1, 2, \cdots$ **do**
4:    Set $\omega^{(0)} = \frac{1}{\mu^{(0)}}$, and $\sigma^{(0)} = \frac{1}{(\mu^{(0)})^{0.1}}$
5:    **for** $k = 0, 1, 2, \cdots$ **do**
6:      Find $(\mathbf{w}_{sy}^{(k)}, \mathbf{w}_{nn}^{(k)}) \in \mathcal{W}_{sy} \times \mathcal{W}_{nn}$, $(\mathbf{y}_1^{(k)}, \cdots, \mathbf{y}_P^{(k)}) \in \mathcal{Y} \times \cdots \times \mathcal{Y}$, and $\mathbf{s}^{(k)} \in \mathbb{R}_{\geq 0}^P$ s.t.

$$\delta^{(k)} \leftarrow \delta(\mathbf{w}_{sy}^{(k)}, \mathbf{w}_{nn}^{(k)}, \mathbf{y}_1^{(k)}, \cdots, \mathbf{y}_p^{(k)}, \mathbf{s}^{(k)}; \lambda^{(k)}, \mu^{(k)}, \iota^{(k)}) \leq \omega^{(k)};$$

7:      **if** $\left( \sum_{i=1}^P c_i(\mathbf{y}_i^{(k)}, \mathbf{w}_{sy}^{(k)}, \mathbf{w}_{nn}^{(k)}, \iota^{(k)}) + s_i \right) < \sigma^{(k)}$ **then**
8:        **if** $\left( \sum_{i=1}^P c_i(\mathbf{y}_i^{(k)}, \mathbf{w}_{sy}^{(k)}, \mathbf{w}_{nn}^{(k)}, \iota^{(k)}) + s_i \right) < \sigma^*$ and $\delta^{(k)} \leq \omega^*$ **then**
9:          Break with the approximate solution: $\mathbf{w}_{sy}^{(k)}, \mathbf{w}_{nn}^{(k)}, \mathbf{y}_1^{(k)}, \cdots, \mathbf{y}_P^{(k)}, \mathbf{s}^{(k)}$;
10:         $\lambda_i^{(k+1)} \leftarrow \lambda_i^{(k)} + \mu^{(k)} \left( c_i(\mathbf{y}_i^{(k)}, \mathbf{w}_{sy}^{(k)}, \mathbf{w}_{nn}^{(k)}, \iota^{(k)}) + s_i \right)$, $\forall i = 1, \cdots, P$;

          $\mu^{(k+1)} \leftarrow \mu^{(k)}$; $\sigma^{(k+1)} \leftarrow \frac{\sigma^{(k)}}{(\mu^{(k+1)})^{0.9}}$; $\omega^{(k+1)} \leftarrow \frac{\omega^{(k)}}{\mu^{(k+1)}}$;
11:       **else**
12:         $\mu^{(k+1)} \leftarrow 2 \cdot \mu^{(k)}$; $\lambda_i^{(k+1)} \leftarrow \lambda_i^{(k)}$, $\forall i = 1, \cdots, P$;
13:         $\sigma^{(k+1)} \leftarrow \frac{1}{(\mu^{(k+1)})^{0.1}}$; $\omega^{(k+1)} \leftarrow \frac{1}{\mu^{(k+1)}}$;
14:     **if** Stopping criterion satisfied **then**
15:       Stop with the approximate solution: $\mathbf{w}_{sy}^{(k)}, \mathbf{w}_{nn}^{(k)}, \mathbf{y}_1^{(k)}, \cdots, \mathbf{y}_P^{(k)}, \mathbf{s}^{(k)}$;
16:     $\mu^{(0)} \leftarrow \mu^{(k)}$; $\lambda_i^{(0)} \leftarrow \lambda_i^{(k)}$, $\forall i = 1, \cdots, P$;
17:     $\iota^{(t+1)} \leftarrow \frac{1}{2} \cdot \iota^{(t)}$;

---

As stated in the main paper, each instance of (11) is optimized using the bound constrained augmented Lagrangian algorithm, Algorithm 17.4 from Nocedal & Wright (2006). This algorithm is applied in lines 4 through 16 in Algorithm 2. The algorithm iteratively finds approximate minimizers of the problem's augmented Lagrangian, (12), for a fixed setting of the penalty parameters using randomized incremental gradient descent, line 6 in Algorithm 2. Specifically, gradient descent is

applied to find an approximate minimizer of (12) satisfying the following stopping criterion:

$$\delta(\mathbf{w}_{sy}, \mathbf{w}_{nn}, \mathbf{y}_1, \cdots, \mathbf{y}_p, \mathbf{s}; \lambda, \mu, \iota) :=$$
$$\left\| \mathbf{w}_{sy} - \Pi\left(\mathbf{w}_{sy} - \nabla_{\mathbf{w}_{sy}} \mathcal{L}_A\right) \right\| + \left\| \mathbf{w}_{nn} - \Pi\left(\mathbf{w}_{nn} - \nabla_{\mathbf{w}_{nn}} \mathcal{L}_A\right) \right\|$$
$$+ \sum_{i=1}^{P} \left\| \mathbf{y}_i - \Pi\left(\mathbf{y}_i - \nabla_{\mathbf{y}_i} \mathcal{L}_A\right) \right\| + \left\| \mathbf{s} - \Pi\left(\mathbf{s} - \nabla_{\mathbf{s}} \mathcal{L}_A\right) \right\| \leq \omega, \tag{20}$$

where $\omega > 0$ is a positive tolerance that is updated with the Lagrange variables. Further, note the Lagrangian gradients are evaluated at the iterate specified as arguments of $\delta$. Practically, the parameter movement between an epoch of incremental gradient descent is used to approximate $\delta$.

As stated in the main paper, employing the bound constrained augmented Lagrangian algorithm to solve the instances of (11) ensures fundamental convergence properties of our learning framework. Specifically, theorem 17.2 in Nocedal & Wright (2006) is applicable to Algorithm 2. This theorem states that limit points of the iterate sequence are stationary points of $\|c(\mathbf{y}_1, \cdots, \mathbf{y}_P, \mathbf{w}_{sy}, \mathbf{w}_{nn})) + \mathbf{s}\|^2$ when they are infeasible or, when the LICQ holds and the iterates are feasible, are KKT points of (11).

## C  EXTENDED NEUPSL AND DEEP HINGE-LOSS MARKOV RANDOM FIELDS

In this section, we expand on the smooth formulation of NeuPSL inference and provide proofs for the continuity results presented in Section 5.2.

### C.1  EXTENDED SMOOTH FORMULATION OF INFERENCE

Recall the primal formulation of NeuPSL inference restated below:

$$\underset{\mathbf{y} \in \mathbb{R}^{n_{\mathbf{y}}}}{\arg\min} \; \mathbf{w}_{sy}^T \mathbf{\Phi}(\mathbf{y}, \mathbf{x}_{sy}, \mathbf{g}(\mathbf{x}_{nn}, \mathbf{w}_{nn})) \quad \text{s.t. } \mathbf{y} \in \mathbf{\Omega}(\mathbf{x}_{sy}, \mathbf{g}(\mathbf{x}_{nn}, \mathbf{w}_{nn})). \tag{21}$$

Importantly, note the structure of the deep hinge-loss potentials defining $\mathbf{\Phi}$:

$$\phi_k(\mathbf{y}, \mathbf{x}_{sy}, \mathbf{g}(\mathbf{x}_{nn}, \mathbf{w}_{nn})) := (\max\{\mathbf{a}_{\phi_k, y}^T \mathbf{y} + \mathbf{a}_{\phi_k, \mathbf{x}_{sy}}^T \mathbf{x}_{sy} + \mathbf{a}_{\phi_k, \mathbf{g}}^T \mathbf{g}(\mathbf{x}_{nn}, \mathbf{w}_{nn}) + b_{\phi_k}, 0\})^{p_k}. \tag{22}$$

The LCQP NeuPSL inference formulation is defined using ordered index sets: $\mathbf{I}_S$ for the partitions of squared hinge potentials (indices $k$ which for all $j \in t_k$ the exponent term $p_j = 2$) and $\mathbf{I}_L$ for the partitions of linear hinge potentials (indices $k$ which for all $j \in t_k$ the exponent term $p_j = 1$). With the index sets, we define

$$\mathbf{W}_S := \begin{bmatrix} w_{\mathbf{I}_S[1]} \mathbf{I} & 0 & \cdots & 0 \\ 0 & w_{\mathbf{I}_S[2]} \mathbf{I} & & \\ \vdots & & \ddots & \end{bmatrix} \quad \text{and} \quad \mathbf{w}_L := \begin{bmatrix} w_{\mathbf{I}_L[1]} \mathbf{1} \\ w_{\mathbf{I}_L[2]} \mathbf{1} \\ \vdots \end{bmatrix} \tag{23}$$

Let $m_S := |\cup_{\mathbf{I}_S} t_k|$ and $m_L := |\cup_{\mathbf{I}_L} t_k|$, be the total number of squared and linear hinge potentials, respectively, and define slack variables $\mathbf{s}_S := [s_j]_{j=1}^{m_S}$ and $\mathbf{s}_L := [s_j]_{j=1}^{m_L}$ for each of the squared and linear hinge potentials, respectively. NeuPSL inference is equivalent to the following LCQP:

$$\min_{\mathbf{y} \in [0,1]^{n_y}, \mathbf{s}_S \in \mathbb{R}^{m_S}, \mathbf{s}_\mathbf{H} \in \mathbb{R}_+^{m_L}} \mathbf{s}_S^T \mathbf{W}_S \mathbf{s}_S + \mathbf{w}_L^T \mathbf{s}_L \tag{24a}$$

$$\text{s.t.} \quad \mathbf{a}_{c_i, \mathbf{y}}^T \mathbf{y} + \mathbf{a}_{c_i, \mathbf{x}_{sy}}^T \mathbf{x}_{sy} + \mathbf{a}_{c_i, \mathbf{g}}^T \mathbf{g}(\mathbf{x}_{nn}, \mathbf{w}_{nn}) + b_{c_i} \leq 0 \quad \forall i = 1, \ldots, q, \tag{24b}$$

$$\mathbf{a}_{\phi_j, \mathbf{y}}^T \mathbf{y} + \mathbf{a}_{\phi_j, \mathbf{x}_{sy}}^T \mathbf{x}_{sy} + \mathbf{a}_{\phi_j, \mathbf{g}}^T \mathbf{g}(\mathbf{x}_{nn}, \mathbf{w}_{nn}) + b_{\phi_j} - s_j \leq 0 \quad \forall j \in I_S \cup I_L. \tag{24c}$$

We ensure strong convexity by adding a square regularization with parameter $\epsilon$ to the objective. Let the bound constraints on $\mathbf{y}$ and $\mathbf{s}_L$ and linear inequalities in the LCQP be captured by the $(2 \cdot n_y + q + m_S + 2 \cdot m_L) \times (n_y + m_S + m_L)$ matrix $\mathbf{A}$ and $(2 \cdot n_y + q + m_S + 2 \cdot m_L)$ dimensional vector $\mathbf{b}(\mathbf{x}_{sy}, \mathbf{g}(\mathbf{x}_{nn}, \mathbf{w}_{nn}))$. More formally, $\mathbf{A} := [a_{ij}]$ where $a_{ij}$ is the coefficient of a decision

variable in the implicit and explicit constraints in the formulation above:

$$
a_{i,j} := \begin{cases}
0 & (i \leq q) \wedge (j \leq m_S + m_L) \\
\mathbf{a}_{c_i,\mathbf{y}}[j - (m_S + m_L)] & (i \leq q) \wedge (j > m_S + m_L) \\
0 & (q < i \leq q + m_S + m_L) \wedge (j \leq m_S + m_L) \wedge (j \neq i - q) \\
-1 & (q < i \leq q + m_S + m_L) \wedge (j \leq m_S + m_L) \wedge (j = i - q) \\
\mathbf{a}_{\phi_{i-q},\mathbf{y}}[j - (m_S + m_L)] & (q < i \leq q + m_S + m_L) \wedge (j > m_S + m_L) \\
0 & (q + m_S + m_L < i \leq q + m_S + 2 \cdot m_L + n_y) \\
& \wedge (j \neq i - (q + m_L)) \\
-1 & (q + m_S + m_L < i \leq q + m_S + 2 \cdot m_L + n_y) \\
& \wedge (j = i - (q + m_L)) \\
0 & (q + m_S + 2 \cdot m_L + n_y < i \leq q + m_S + 2 \cdot m_L + 2 \cdot n_y) \\
& \wedge (j \neq i - (q + m_S + m_L)) \\
1 & (q + m_S + 2 \cdot m_L + n_y < i \leq q + m_S + 2 \cdot m_L + 2 \cdot n_y) \\
& \wedge (j = i - (q + m_S + m_L))
\end{cases} .
$$

(25)

Furthermore, $\mathbf{b}(\mathbf{x}_{sy}, \mathbf{g}(\mathbf{x}_{nn}, \mathbf{w}_{nn})) = [b_i(\mathbf{x}_{sy}, \mathbf{g}(\mathbf{x}_{nn}, \mathbf{w}_{nn}))]$ is the vector of constants corresponding to each constraint in the formulation above:

$$
b_i(\mathbf{x}_{sy}, \mathbf{g}(\mathbf{x}_{nn}, \mathbf{w}_{nn}))
$$

(26)

$$
:= \begin{cases}
\mathbf{a}_{c_i,\mathbf{x}_{sy}}^T \mathbf{x}_{sy} + \mathbf{a}_{c_i,\mathbf{g}}^T \mathbf{g}(\mathbf{x}_{nn}, \mathbf{w}_{nn}) + b_{c_i} & i \leq q \\
\mathbf{a}_{\phi_{i-q},\mathbf{x}_{sy}}^T \mathbf{x}_{sy} + \mathbf{a}_{\phi_{i-q},\mathbf{g}}^T \mathbf{g}(\mathbf{x}_{nn}, \mathbf{w}_{nn}) + b_{\phi_{i-q}} & q < i \leq q + m_S + m_L \\
0 & q + m_S + m_L < i \\
& \quad \leq q + m_S + 2 \cdot m_L + n_y \\
-1 & q + m_S + 2 \cdot m_L + n_y < i \\
& \quad \leq q + m_S + 2 \cdot m_L + 2 \cdot n_y
\end{cases} .
$$

(27)

Note that $\mathbf{b}(\mathbf{x}_{sy}, \mathbf{g}(\mathbf{x}_{nn}, \mathbf{w}_{nn}))$ is a linear function of the neural network outputs, hence, if $\mathbf{g}(\mathbf{x}_{nn}, \mathbf{w}_{nn})$ is a smooth function of the neural parameters, then $\mathbf{b}(\mathbf{x}_{sy}, \mathbf{g}(\mathbf{x}_{nn}, \mathbf{w}_{nn}))$ is also smooth.

With this notation, the regularized inference problem is:

$$
V(\mathbf{w}_{sy}, \mathbf{b}(\mathbf{x}_{sy}, \mathbf{g}(\mathbf{x}_{nn}, \mathbf{w}_{nn}))) := \min_{\mathbf{y}, \mathbf{s}_S, \mathbf{s}_H} \begin{bmatrix} \mathbf{s}_S \\ \mathbf{s}_L \\ \mathbf{y} \end{bmatrix}^T \begin{bmatrix} \mathbf{W}_S + \epsilon I & 0 & 0 \\ 0 & \epsilon I & 0 \\ 0 & 0 & \epsilon I \end{bmatrix} \begin{bmatrix} \mathbf{s}_S \\ \mathbf{s}_L \\ \mathbf{y} \end{bmatrix} + \begin{bmatrix} 0 \\ \mathbf{w}_L \\ 0 \end{bmatrix}^T \begin{bmatrix} \mathbf{s}_S \\ \mathbf{s}_L \\ \mathbf{y} \end{bmatrix}
$$

$$
\text{s.t.} \quad \mathbf{A} \begin{bmatrix} \mathbf{s}_S \\ \mathbf{s}_L \\ \mathbf{y} \end{bmatrix} + \mathbf{b}(\mathbf{x}_{sy}, \mathbf{g}(\mathbf{x}_{nn}, \mathbf{w}_{nn})) \leq 0.
$$

(28)

For ease of notation, let

$$
D(\mathbf{w}_{sy}) := \begin{bmatrix} \mathbf{W}_S & 0 & 0 \\ 0 & 0 & 0 \\ 0 & 0 & 0 \end{bmatrix}, \quad \mathbf{c}(\mathbf{w}_{sy}) := \begin{bmatrix} 0 \\ \mathbf{w}_L \\ 0 \end{bmatrix}, \quad \nu := \begin{bmatrix} \mathbf{s}_S \\ \mathbf{s}_L \\ \mathbf{y} \end{bmatrix}.
$$

(29)

Then the regularized primal LCQP MAP inference problem is concisely expressed as

$$
\min_{\nu \in \mathbb{R}^{n_{\mathbf{y}} + m_S + m_L}} \nu^T (\mathbf{D}(\mathbf{w}_{sy}) + \epsilon \mathbf{I}) \nu + \mathbf{c}(\mathbf{w}_{sy})^T \nu
$$

(30)

$$
\text{s.t.} \quad \mathbf{A}\nu + \mathbf{b}(\mathbf{x}_{sy}, \mathbf{g}(\mathbf{x}_{nn}, \mathbf{w}_{nn})) \leq 0.
$$

By Slater's constraint qualification, we have strong-duality when there is a feasible solution. In this case, an optimal solution to the dual problem yields an optimal solution to the primal problem. The

Lagrange dual problem of (30) is

$$
\begin{aligned}
\underset{\mu \geq 0}{\arg\max} \ \underset{\nu \in \mathbb{R}^{n_{\mathbf{y}} + m_S + m_L}}{\min} \ & \nu^T (\mathbf{D}(\mathbf{w}_{sy}) + \epsilon \mathbf{I}) \nu + \mathbf{c}(\mathbf{w}_{sy})^T \nu + \mu^T (\mathbf{A}\nu + \mathbf{b}(\mathbf{x}_{sy}, \mathbf{g}(\mathbf{x}_{nn}, \mathbf{w}_{nn}))) \\
= \underset{\mu \geq 0}{\arg\max} \ & -\frac{1}{4} \mu^T \mathbf{A} (\mathbf{D}(\mathbf{w}_{sy}) + \epsilon \mathbf{I})^{-1} \mathbf{A}^T \mu \\
& - \frac{1}{2} (\mathbf{A}(\mathbf{D}(\mathbf{w}_{sy}) + \epsilon \mathbf{I})^{-1} \mathbf{c}(\mathbf{w}_{sy}) - 2\mathbf{b}(\mathbf{x}_{sy}, \mathbf{g}(\mathbf{x}_{nn}, \mathbf{w}_{nn})))^T \mu
\end{aligned}
\tag{31}
$$

where $\mu = [\mu_i]_{i=1}^{n_\mu}$ are the Lagrange dual variables. For later reference, denote the negative of the Lagrange dual function of MAP inference as:

$$
\begin{aligned}
& h(\mu; \mathbf{w}_{sy}, \mathbf{b}(\mathbf{x}_{sy}, \mathbf{g}(\mathbf{x}_{nn}, \mathbf{w}_{nn}))) \\
& := \frac{1}{4} \mu^T \mathbf{A} (\mathbf{D}(\mathbf{w}_{sy}) + \epsilon \mathbf{I})^{-1} \mathbf{A}^T \mu + \frac{1}{2} (\mathbf{A}(\mathbf{D}(\mathbf{w}_{sy}) + \epsilon \mathbf{I})^{-1} \mathbf{c}(\mathbf{w}_{sy}) - 2\mathbf{b}(\mathbf{x}_{sy}, \mathbf{g}(\mathbf{x}_{nn}, \mathbf{w}_{nn})))^T \mu.
\end{aligned}
\tag{32}
$$

The dual LCQP has more decision variables but is only over non-negativity constraints rather than the complex polyhedron feasible set. The dual-to-primal variable translation is:

$$
\nu = -\frac{1}{2} (\mathbf{D}(\mathbf{w}_{sy}) + \epsilon \mathbf{I})^{-1} (\mathbf{A}^T \mu + \mathbf{c}(\mathbf{w}_{sy}))
\tag{33}
$$

As $(\mathbf{D}(\mathbf{w}_{sy}) + \epsilon \mathbf{I})$ is diagonal, it is easy to invert and hence it is practical to work in the dual space to obtain a solution to the primal problem.

## C.2 EXTENDED CONTINUITY OF INFERENCE

We now provide background on sensitivity analysis that we then apply in our proofs on the continuity properties of NeuPSL inference.

### C.2.1 BACKGROUND

**Theorem C.1** (Boyd & Vandenberghe (2004) p. 81)**.** *If for each* $\mathbf{y} \in \mathcal{A}$, $f(\mathbf{x}, \mathbf{y})$ *is convex in* $\mathbf{x}$ *then the function*

$$
g(\mathbf{x}) := \sup_{\mathbf{y} \in \mathcal{A}} f(\mathbf{x}, \mathbf{y})
\tag{34}
$$

*is convex in* $\mathbf{x}$.

**Theorem C.2** (Boyd & Vandenberghe (2004) p. 81)**.** *If for each* $\mathbf{y} \in \mathcal{A}$, $f(\mathbf{x}, \mathbf{y})$ *is concave in* $\mathbf{x}$ *then the function*

$$
g(\mathbf{x}) := \inf_{\mathbf{y} \in \mathcal{A}} f(\mathbf{x}, \mathbf{y})
\tag{35}
$$

*is concave in* $\mathbf{x}$.

**Definition C.3** (Convex Subgradient: Boyd & Vandenberghe (2004) and Shalev-Shwartz (2012))**.** Consider a convex function $f : \mathbb{R}^n \to [-\infty, \infty]$ and a point $\overline{\mathbf{x}}$ with $f(\overline{\mathbf{x}})$ finite. For a vector $\mathbf{v} \in \mathbf{R}^n$, one says that $\mathbf{v}$ is a (convex) subgradient of $f$ at $\overline{\mathbf{x}}$, written $\mathbf{v} \in \partial f(\overline{\mathbf{x}})$, iff

$$
f(\mathbf{x}) \geq f(\overline{\mathbf{x}}) + <\mathbf{v}, \mathbf{x} - \overline{\mathbf{x}}>, \quad \forall \mathbf{x} \in \mathbf{R}^n.
\tag{36}
$$

**Definition C.4** (Closedness: Bertsekas (2009))**.** If the epigraph of a function $f : \mathbb{R}^n \to [-\infty, \infty]$ is a closed set, we say that $f$ is a closed function.

**Definition C.5** (Lower Semicontinuity: Bertsekas (2009))**.** The function $f : \mathbb{R}^n \to [-\infty, \infty]$ is *lower semicontinuous* (lsc) at a point $\overline{\mathbf{x}} \in \mathbb{R}^n$ if

$$
f(\overline{\mathbf{x}}) \leq \liminf_{k \to \infty} f(\mathbf{x}_k),
\tag{37}
$$

for every sequence $\{\mathbf{x}_k\} \subset \mathbb{R}^n$ with $\mathbf{x}_k \to \overline{\mathbf{x}}$. We say $f$ is *lsc* if it is lsc at each $\overline{\mathbf{x}}$ in its domain.

**Theorem C.6** (Closedness and Semicontinuity: Bertsekas (2009) Proposition 1.1.2.)**.** *For a function* $f : \mathbb{R}^n \to [-\infty, \infty]$, *the following are equivalent:*

1. *The level set $V_\gamma = \{\mathbf{x} \mid f(\mathbf{x}) \leq \gamma\}$ is closed for every scalar $\gamma$.*

2. *$f$ is lsc.*

3. *$f$ is closed.*

The following definition and theorem are from Rockafellar & Wets (1997) and they generalize the notion of subgradients to non-convex functions and the chain rule of differentiation, respectively. For complete statements see Rockafellar & Wets (1997) Rockafellar & Wets (1997).

**Definition C.7** (Regular Subgradient: Rockafellar & Wets (1997) Definition 8.3). Consider a function $f : \mathbf{R}^n \to [-\infty, \infty]$ and a point $\overline{\mathbf{x}}$ with $f(\overline{\mathbf{x}})$ finite. For a vector $\mathbf{v} \in \mathbf{R}^n$, one says that $\mathbf{v}$ is a regular subgradient of $f$ at $\overline{\mathbf{x}}$, written $\mathbf{v} \in \hat{\partial} f(\overline{\mathbf{x}})$, iff

$$f(\mathbf{x}) \geq f(\overline{\mathbf{x}}) + \langle \mathbf{v}, \mathbf{x} - \overline{\mathbf{x}} \rangle + \mathrm{o}(\mathbf{x} - \overline{\mathbf{x}}), \quad \forall \mathbf{x} \in \mathbf{R}^n, \tag{38}$$

where the $\mathrm{o}(t)$ notation indicates a term with the property that

$$\lim_{t \to 0} \frac{\mathrm{o}(t)}{t} = 0. \tag{39}$$

The relation of the regular subgradient defined above and the more familiar convex subgradient is the addition of the $o(\mathbf{x} - \overline{\mathbf{x}})$ term. Evidently, a convex subgradient is a regular subgradient.

**Theorem C.8** (Chain Rule for Regular Subgradients: Rockafellar & Wets (1997) Theorem 10.6). *Suppose $f(\mathbf{x}) = g(F(\mathbf{x}))$ for a proper, lsc function $g : \mathbb{R}^m \to [-\infty, \infty]$ and a smooth mapping $F : \mathbb{R}^n \to \mathbb{R}^m$. Then at any point $\overline{\mathbf{x}} \in dom\, f = F^{-1}(dom\, g)$ one has*

$$\hat{\partial} f(\overline{\mathbf{x}}) \supset \nabla F(\overline{\mathbf{x}})^T \hat{\partial} g(F(\overline{\mathbf{x}})), \tag{40}$$

*where $\nabla F(\overline{\mathbf{x}})^T$ is the Jacobian of $F$ at $\overline{\mathbf{x}}$.*

**Theorem C.9** (Danskin's Theorem: Danskin (1966) and Bertsekas (1971) Proposition A.22). *Suppose $\mathcal{Z} \subseteq \mathbb{R}^m$ is a compact set and $g(\mathbf{x}, \mathbf{z}) : \mathbb{R}^n \times \mathcal{Z} \to (-\infty, \infty]$ is a function. Suppose $g(\cdot, \mathbf{z}) : \mathbb{R}^n \to \mathbb{R}$ is closed proper convex function for every $\mathbf{z} \in \mathcal{Z}$. Further, define the function $f : \mathbb{R}^n \to \mathbb{R}$ such that*

$$f(\mathbf{x}) := \max_{\mathbf{z} \in \mathcal{Z}} g(\mathbf{x}, \mathbf{z}).$$

*Suppose $f$ is finite somewhere. Moreover, let $\mathcal{X} := int(dom f)$, i.e., the interior of the set of points in $\mathbb{R}^n$ such that $f$ is finite. Suppose $g$ is continuous on $\mathcal{X} \times \mathcal{Z}$. Further, define the set of maximizing points of $g(\mathbf{x}, \cdot)$ for each $\mathbf{x}$*

$$Z(\mathbf{x}) = \arg \max_{\mathbf{z} \in \mathcal{Z}} g(\mathbf{x}, \mathbf{z}).$$

*Then the following properties of $f$ hold.*

1. *The function $f(\mathbf{x})$ is a closed proper convex function.*

2. *For every $\mathbf{x} \in \mathcal{X}$,*

$$\partial f(\mathbf{x}) = conv\, \{\partial_{\mathbf{x}} g(\mathbf{x}, \mathbf{z}) \mid \mathbf{z} \in Z(\mathbf{x})\}. \tag{41}$$

**Corollary C.10.** *Assume the conditions for Danskin's Theorem above hold. For every $\mathbf{x} \in \mathcal{X}$, if $Z(\mathbf{x})$ consists of a unique point, call it $\mathbf{z}^*$, and $g(\cdot, \mathbf{z}^*)$ is differentiable at $\mathbf{x}$, then $f(\cdot)$ is differentiable at $\mathbf{x}$, and*

$$\nabla f(\mathbf{x}) := \nabla_{\mathbf{x}} g(\mathbf{x}, \mathbf{z}^*). \tag{42}$$

**Theorem C.11** (Bonnans & Shapiro (1998) Theorem 4.2, Rockafellar (1974) p. 41). *Let $\mathbf{X}$ and $\mathbf{U}$ be Banach spaces. Let $\mathbf{K}$ be a closed convex cone in the Banach space $\mathbf{U}$. Let $G : \mathbf{X} \to \mathbf{U}$ be a convex mapping with respect to the cone $\mathbf{C} := -\mathbf{K}$ and $f : \mathbf{X} \to (-\infty, \infty]$ be a (possibly infinite-valued) convex function. Consider the following convex program and its optimal value function:*

$$v_P(\mathbf{u}) := \min_{\mathbf{x} \in \mathbf{X}} f(\mathbf{x}) \qquad\qquad (P)$$

$$s.t. \quad G(\mathbf{x}) + \mathbf{u} \in \mathbf{K}.$$

*Moreover, consider the (Lagrangian) dual of the program:*

$$v_D(\mathbf{u}) := \max_{\lambda \in \mathbf{K}^-} \min_{\mathbf{x} \in \mathbf{X}} f(\mathbf{x}) + \lambda^T(G(\mathbf{x}) + \mathbf{u}) \qquad (D)$$

*Suppose $v_P(\mathbf{0})$ is finite. Further, suppose the feasible set of the program is nonempty for all $\mathbf{u}$ in a neighborhood of $\mathbf{0}$, i.e.,*

$$\mathbf{0} \in int\{G(\mathbf{X}) - \mathbf{K}\}. \qquad (43)$$

*Then,*

1. *There is no primal dual gap at $u = 0$, i.e., $v_P(0) = v_D(0)$.*

2. *The set, $\Lambda_0$, of optimal solutions to the dual problem with $\mathbf{u} = 0$ is non-empty and bounded.*

3. *The optimal value function $v_P(\mathbf{u})$ is continuous at $\mathbf{u} = 0$ and $\partial v_P(\mathbf{0}) = \Lambda_0$.*

**Theorem C.12** (Bonnans & Shapiro (2000) Proposition 4.3.2). *Consider two optimization problems over a non-empty feasible set $\mathbf{\Omega}$:*

$$\min_{\mathbf{x} \in \mathbf{\Omega}} f_1(\mathbf{x}) \qquad and \qquad \min_{\mathbf{x} \in \mathbf{\Omega}} f_2(\mathbf{x}) \qquad (44)$$

*where $f_1, f_2 : \mathcal{X} \to \mathbb{R}$. Suppose $f_1$ has a non-empty set $\mathbf{S}$ of optimal solutions over $\mathbf{\Omega}$. Suppose the second order growth condition holds for $\mathbf{S}$, i.e., there exists a neighborhood $\mathcal{N}$ of $\mathbf{S}$ and a constant $\alpha > 0$ such that*

$$f_1(\mathbf{x}) \geq f_1(\mathbf{S}) + \alpha(dist(\mathbf{x}, \mathbf{S}))^2, \qquad \forall \mathbf{x} \in \mathbf{\Omega} \cap \mathcal{N}, \qquad (45)$$

*where $f_1(\mathbf{S}) := \inf_{\mathbf{x} \in \mathbf{\Omega}} f_1(\mathbf{x})$. Define the difference function:*

$$\Delta(\mathbf{x}) := f_2(\mathbf{x}) - f_1(\mathbf{x}). \qquad (46)$$

*Suppose $\Delta(\mathbf{x})$ is L-Lipschitz continuous on $\mathbf{\Omega} \cap \mathcal{N}$. Let $\mathbf{x}^* \in \mathcal{N}$ be an $\delta$-solution to the problem of minimizing $f_2(\mathbf{x})$ over $\mathbf{\Omega}$. Then*

$$dist(\mathbf{x}^*, \mathbf{S}) \leq \frac{L}{\alpha} + \sqrt{\frac{\delta}{\alpha}}. \qquad (47)$$

### C.2.2 PROOFS

We provide proofs of theorems presented in the main paper and restated here for completeness.

***Theorem 5.2***. Suppose for any setting of $\mathbf{w}_{nn} \in \mathbb{R}^{n_g}$ there is a feasible solution to NeuPSL inference (16). Further, suppose $\epsilon > 0$, $\mathbf{w}_{sy} \in \mathbb{R}^r_+$, and $\mathbf{w}_{nn} \in \mathbb{R}^{n_g}$. Then:

- The minimizer of (16), $\mathbf{y}^*(\mathbf{w}_{sy}, \mathbf{w}_{nn})$, is a $O(1/\epsilon)$ Lipschitz continuous function of $\mathbf{w}_{sy}$.
- $V(\mathbf{w}_{sy}, \mathbf{b}(\mathbf{x}_{sy}, \mathbf{g}(\mathbf{x}_{nn}, \mathbf{w}_{nn})))$, is concave over $\mathbf{w}_{sy}$ and convex over $\mathbf{b}(\mathbf{x}_{sy}, \mathbf{g}(\mathbf{x}_{nn}, \mathbf{w}_{nn}))$.
- $V(\mathbf{w}_{sy}, \mathbf{b}(\mathbf{x}_{sy}, \mathbf{g}(\mathbf{x}_{nn}, \mathbf{w}_{nn})))$ is differentiable with respect to $\mathbf{w}_{sy}$. Moreover,

$$\nabla_{\mathbf{w}_{sy}} V(\mathbf{w}_{sy}, \mathbf{b}(\mathbf{x}_{sy}, \mathbf{g}(\mathbf{x}_{nn}, \mathbf{w}_{nn}))) = \mathbf{\Phi}(\mathbf{y}^*(\mathbf{w}_{sy}, \mathbf{w}_{nn}), \mathbf{x}_{sy}, \mathbf{g}(\mathbf{x}_{nn}, \mathbf{w}_{nn})).$$

  Furthermore, $\nabla_{\mathbf{w}_{sy}} V(\mathbf{w}_{sy}, \mathbf{b}(\mathbf{x}_{sy}, \mathbf{g}(\mathbf{x}_{nn}, \mathbf{w}_{nn})))$ is Lipschitz continuous over $\mathbf{w}_{sy}$.
- If there is a feasible point $\nu$ strictly satisfying the $i'th$ inequality constraint of (16), i.e., $\mathbf{A}[i]\nu + \mathbf{b}(\mathbf{x}_{sy}, \mathbf{g}(\mathbf{x}_{nn}, \mathbf{w}_{nn}))[i] < 0$, then $V(\mathbf{w}_{sy}, \mathbf{b}(\mathbf{x}_{sy}, \mathbf{g}(\mathbf{x}_{nn}, \mathbf{w}_{nn})))$ is subdifferentiable with respect to the $i'th$ constraint constant $\mathbf{b}(\mathbf{x}_{sy}, \mathbf{g}(\mathbf{x}_{nn}, \mathbf{w}_{nn}))[i]$. Moreover,

$$\partial_{\mathbf{b}[i]} V(\mathbf{w}_{sy}, \mathbf{b}(\mathbf{x}_{sy}, \mathbf{g}(\mathbf{x}_{nn}, \mathbf{w}_{nn}))) = \{\mu^*[i] \,|\, \mu^* \in \underset{\mu \in \mathbb{R}_{\geq 0}^{2 \cdot n_{\mathbf{y}} + m + q}}{\arg\min} h(\mu; \mathbf{w}_{sy}, \mathbf{b}(\mathbf{x}_{sy}, \mathbf{g}(\mathbf{x}_{nn}, \mathbf{w}_{nn})))\}.$$

  Furthermore, if $\mathbf{g}(\mathbf{x}_{nn}, \mathbf{w}_{nn})$ is a smooth function of $\mathbf{w}_{nn}$, then so is $\mathbf{b}(\mathbf{x}_{sy}, \mathbf{g}(\mathbf{x}_{nn}, \mathbf{w}_{nn}))$, and the set of regular subgradients of $V(\mathbf{w}_{sy}, \mathbf{b}(\mathbf{x}_{sy}, \mathbf{g}(\mathbf{x}_{nn}, \mathbf{w}_{nn})))$ is:

$$\hat{\partial}_{\mathbf{w}_{nn}} V(\mathbf{w}_{sy}, \mathbf{b}(\mathbf{x}_{sy}, \mathbf{g}(\mathbf{x}_{nn}, \mathbf{w}_{nn}))) \qquad (48)$$
$$\supset \nabla_{\mathbf{w}_{nn}} \mathbf{b}(\mathbf{x}_{sy}, \mathbf{g}(\mathbf{x}_{nn}, \mathbf{w}_{nn}))^T \partial_{\mathbf{b}} V(\mathbf{w}_{sy}, \mathbf{b}(\mathbf{x}_{sy}, \mathbf{g}(\mathbf{x}_{nn}, \mathbf{w}_{nn}))).$$

***Proof of Theorem 5.2.*** We first show the minimizer of the LCQP formulation of NeuPSL inference, $\nu^*$, with $\epsilon > 0$, $\mathbf{w}_{sy} \in \mathbb{R}_+^r$, and $\mathbf{w}_{nn} \in \mathbb{R}^{n_g}$ is a Lipschitz continuous function of $\mathbf{w}_{sy}$. Suppose $\epsilon > 0$. To show continuity over $\mathbf{w}_{sy} \in \mathbb{R}_+^r$, first note the matrix $(\mathbf{D} + \epsilon\mathbf{I})$ is positive definite and the primal inference problem (17) is an $\epsilon$-strongly convex LCQP with a unique minimizer denoted by $\nu^*(\mathbf{w}_{sy}, \mathbf{w}_{nn})$. We leverage the Lipschitz stability result for optimal values of constrained problems from Bonnans & Shapiro (2000) and presented here in Theorem C.12. Define the primal objective as an explicit function of the weights:

$$f(\nu, \mathbf{w}_{sy}, \mathbf{w}_{nn}) := \nu^T(\mathbf{D}(\mathbf{w}_{sy}) + \epsilon\mathbf{I})\nu + \mathbf{c}^T(\mathbf{w}_{sy})\nu \tag{49}$$

Note that the solution $\nu^* = \begin{bmatrix} \mathbf{s}_S^* \\ \mathbf{s}_L^* \\ \mathbf{y}^* \end{bmatrix}$ will always be bounded, since from (24c) in LCQP we always have for all $j \in I_S \cup I_L$,

$$0 \leq s_j^* = \max(\mathbf{a}_{\phi_k,y}^T\mathbf{y}^* + \mathbf{a}_{\phi_k,\mathbf{x}_{sy}}^T\mathbf{x}_{sy} + \mathbf{a}_{\phi_k,\mathbf{g}}^T\mathbf{g}(\mathbf{x}_{nn}, \mathbf{w}_{nn}) + b_{\phi_k}, 0) \tag{50}$$

$$\leq \|\mathbf{a}_{\phi_k,y}\| + |\mathbf{a}_{\phi_k,\mathbf{x}_{sy}}^T\mathbf{x}_{sy} + \mathbf{a}_{\phi_k,\mathbf{g}}^T\mathbf{g}(\mathbf{x}_{nn}, \mathbf{w}_{nn}) + b_{\phi_k}|. \tag{51}$$

Thus, setting these trivial upper bounds for $s_j$ will not change the solution of the problem. We can henceforth consider the problem in a bounded domain $\|\nu\| \leq C$ where $C$ does not depend on $\mathbf{w}$'s.

Let $\mathbf{w}_{1,sy}, \mathbf{w}_{2,sy} \in \mathbb{R}_+^r$ and $\mathbf{w}_{nn} \in \mathcal{W}_{nn}$ be arbitrary. As $\epsilon > 0$, $f(\nu, \mathbf{w}_{1,sy}, \mathbf{w}_{nn})$ is strongly convex in $\nu$ and it therefore satisfies the second-order growth condition in $\nu$. Define the difference function:

$$\Delta_{\mathbf{w}_{sy}}(\nu) := f(\nu, \mathbf{w}_{2,sy}, \mathbf{w}_{nn}) - f(\nu, \mathbf{w}_{1,sy}, \mathbf{w}_{nn}) \tag{52}$$

$$= \nu^T(\mathbf{D}(\mathbf{w}_{2,sy}) + \epsilon\mathbf{I})\nu + \mathbf{c}^T(\mathbf{w}_{2,sy})\nu - \left(\nu^T(\mathbf{D}(\mathbf{w}_{1,sy}) + \epsilon\mathbf{I})\nu + \mathbf{c}^T(\mathbf{w}_{1,sy})\nu\right) \tag{53}$$

$$= \nu^T(\mathbf{D}(\mathbf{w}_{2,sy}) - \mathbf{D}(\mathbf{w}_{1,sy}))\nu + (\mathbf{c}(\mathbf{w}_{2,sy}) - \mathbf{c}(\mathbf{w}_{1,sy}))^T\nu. \tag{54}$$

The difference function $\Delta_{\mathbf{w}_{sy}}(\nu)$ over $\mathcal{N}$ has a finitely bounded gradient:

$$\|\nabla\Delta_{\mathbf{w}_{sy}}(\nu)\|_2 = \left\|2(\mathbf{D}(\mathbf{w}_{2,sy}) - \mathbf{D}(\mathbf{w}_{1,sy}))\nu + \mathbf{c}(\mathbf{w}_{2,sy}) - \mathbf{c}(\mathbf{w}_{1,sy})\right\|_2 \tag{55}$$

$$\leq \|\mathbf{c}(\mathbf{w}_{2,sy}) - \mathbf{c}(\mathbf{w}_{1,sy})\|_2 + 2\|(\mathbf{D}(\mathbf{w}_{2,sy}) - \mathbf{D}(\mathbf{w}_{1,sy}))\nu\|_2 \tag{56}$$

$$\leq \|\mathbf{w}_{2,sy} - \mathbf{w}_{1,sy}\|_2 + 2\|\mathbf{w}_{2,sy} - \mathbf{w}_{1,sy}\|_2 \|\nu\|_2 \tag{57}$$

$$\leq \|\mathbf{w}_{2,sy} - \mathbf{w}_{1,sy}\|_2(1 + 2C) =: L_\mathcal{N}(\mathbf{w}_{1,sy}, \mathbf{w}_{2,sy}). \tag{58}$$

Thus, the distance function, $\Delta_{\mathbf{w}_{sy}}(\nu)$ is $L_\mathcal{N}(\mathbf{w}_{1,sy}, \mathbf{w}_{2,sy})$-Lipschitz continuous over $\mathcal{N}$. Therefore, by Bonnans & Shapiro (2000) (Theorem C.12), the distance between $\nu^*(\mathbf{w}_{1,sy}, \mathbf{w}_{nn})$ and $\nu^*(\mathbf{w}_{2,sy}, \mathbf{w}_{nn})$ is bounded above:

$$\|\nu^*(\mathbf{w}_{2,sy}, \mathbf{w}_{nn}) - \nu^*(\mathbf{w}_{1,sy}, \mathbf{w}_{nn})\|_2 \leq \frac{L_\mathcal{N}(\mathbf{w}_{1,sy}, \mathbf{w}_{2,sy})}{\epsilon} = \frac{(1 + 2C)}{\epsilon}\|\mathbf{w}_{2,sy} - \mathbf{w}_{1,sy}\|_2. \tag{59}$$

Therefore, the function $\nu^*(\mathbf{w}_{sy}, \mathbf{w}_{nn})$ is $O(1/\epsilon)$-Lipschitz continuous in $\mathbf{w}_{sy}$ for any $\mathbf{w}_{nn}$.

Next, we prove curvature properties of the value-function with respect to the weights. Observe NeuPSL inference is an infimum over a set of functions that are concave (affine) in $\mathbf{w}_{sy}$. Therefore, by Theorem C.2, we have that $V(\mathbf{w}_{sy}, \mathbf{b}(\mathbf{x}_{sy}, \mathbf{g}(\mathbf{x}_{nn}, \mathbf{w}_{nn})))$ is concave in $\mathbf{w}_{sy}$.

We use a similar argument to show $V(\mathbf{w}_{sy}, \mathbf{b}(\mathbf{x}_{sy}, \mathbf{g}(\mathbf{x}_{nn}, \mathbf{w}_{nn})))$ is convex in the constraint constants, $\mathbf{b}(\mathbf{x}_{sy}, \mathbf{g}(\mathbf{x}_{nn}, \mathbf{w}_{nn}))$. Assuming for any setting of the neural weights, $\mathbf{w}_{nn} \in \mathbb{R}^{n_g}$, there is a feasible solution to the NeuPSL inference problem, then (16) satisfies the conditions for Slater's constraint qualification. Therefore, strong duality holds, i.e., $V(\mathbf{w}_{sy}, \mathbf{b}(\mathbf{x}_{sy}, \mathbf{g}(\mathbf{x}_{nn}, \mathbf{w}_{nn})))$ is equal to the optimal value of the dual inference problem (31). Observe that the dual NeuPSL inference problem is a supremum over a set of functions convex (affine) in $\mathbf{b}(\mathbf{x}_{sy}, \mathbf{g}(\mathbf{x}_{nn}, \mathbf{w}_{nn}))$. Therefore, by Theorem C.1, we have that $V(\mathbf{w}_{sy}, \mathbf{b}(\mathbf{x}_{sy}, \mathbf{g}(\mathbf{x}_{nn}, \mathbf{w}_{nn})))$ is convex in $\mathbf{b}(\mathbf{x}_{sy}, \mathbf{g}(\mathbf{x}_{nn}, \mathbf{w}_{nn}))$.

We can additionally prove convexity in $\mathbf{b}$ from first principles. For simplicity we fix other parameters, and write the objective and the value function as $Q(\nu)$ and $V(\mathbf{b})$. Given two values $\mathbf{b}_1$ and

$\mathbf{b}_2$, let corresponding optimal values of (33) be $\nu_1$ and $\nu_2$. Take any $\alpha \in [0, 1]$, note that when $\mathbf{b} = \alpha \mathbf{b}_1 + (1 - \alpha)\mathbf{b}_2$, then $\alpha \nu_1 + (1 - \alpha)\nu_2$ is feasible for this $\mathbf{b}$. Because we take the inf over all $\nu$s, the optimal $\nu$ for this $\mathbf{b}$ might be even smaller. Thus, we have (for convex quadratic objective $Q$) that

$$
\begin{aligned}
V(\alpha b_1 + (1 - \alpha)b_2) &\le Q(\alpha \nu_1 + (1 - \alpha)\nu_2) \\
&\le \alpha Q(\nu_1) + (1 - \alpha)Q(\nu_2) \\
&= \alpha V(b_1) + (1 - \alpha)V(b_2),
\end{aligned}
\tag{60}
$$

which shows that $V$ is convex in $\mathbf{b}$.

Next, we prove (sub)differentiability properties of the value-function. Suppose $\epsilon > 0$. First, we show the optimal value function, $V(\mathbf{w}_{sy}, \mathbf{b}(\mathbf{x}_{sy}, \mathbf{g}(\mathbf{x}_{nn}, \mathbf{w}_{nn})))$, is differentiable with respect to the symbolic weights. Then we show subdifferentiability properties of the optimal value function with respect to the constraint constants. Finally, we apply the Lipschitz continuity of the minimzer result to show the gradient of the optimal value function is Lipschitz continuous with respect to $\mathbf{w}_{sy}$.

Starting with differentiability with respect to the symbolic weights, $\mathbf{w}_{sy}$, note, the optimal value function of the regularized LCQP formulation of NeuPSL inference, (16), is equivalently expressed as the following maximization over a continuous function in the primal target variables, $\mathbf{y}$, the slack variables, $\mathbf{s}_S$ and $\mathbf{s}_L$, and the symbolic weights, $\mathbf{w}_{sy}$:

$$
V(\mathbf{w}_{sy}, \mathbf{b}(\mathbf{x}_{sy}, \mathbf{g}(\mathbf{x}_{nn}, \mathbf{w}_{nn}))) \tag{61}
$$
$$
= -\left( \max_{\mathbf{y}, \mathbf{s_H}, \mathbf{s_L}} - \left( \begin{bmatrix} \mathbf{s}_S \\ \mathbf{s}_L \\ \mathbf{y} \end{bmatrix}^T \begin{bmatrix} \mathbf{W}_S + \epsilon I & 0 & 0 \\ 0 & \epsilon I & 0 \\ 0 & 0 & \epsilon I \end{bmatrix} \begin{bmatrix} \mathbf{s}_S \\ \mathbf{s}_L \\ \mathbf{y} \end{bmatrix} + \begin{bmatrix} 0 \\ \mathbf{w}_L \\ 0 \end{bmatrix}^T \begin{bmatrix} \mathbf{s}_S \\ \mathbf{s}_L \\ \mathbf{y} \end{bmatrix} \right) \right)
$$
$$
\text{s.t.} \quad \mathbf{A} \begin{bmatrix} \mathbf{s}_S \\ \mathbf{s}_L \\ \mathbf{y} \end{bmatrix} + \mathbf{b}(\mathbf{x}_{sy}, \mathbf{g}(\mathbf{x}_{nn}, \mathbf{w}_{nn}) \le 0,
$$

where the matrix $\mathbf{W}_s$ and vector $\mathbf{w}_L$ are functions of the symbolic parameters $\mathbf{w}_{sy}$ as defined in (23). Moreover, the objective above is and convex (affine) in $\mathbf{w}_{sy}$. Additionally, note that the decision variables can be constrained to a compact domain without breaking the equivalence of the formulation. Specifically, the target variables are constrained to the box $[0, 1]^{\mathbf{n}_y}$, while the slack variables are nonnegative and have a trivial upper bound derived from (24c):,

$$
\begin{aligned}
0 \le s_j^* &= \max(\mathbf{a}_{\phi_k, y}^T \mathbf{y}^* + \mathbf{a}_{\phi_k, \mathbf{x}_{sy}}^T \mathbf{x}_{sy} + \mathbf{a}_{\phi_k, \mathbf{g}}^T \mathbf{g}(\mathbf{x}_{nn}, \mathbf{w}_{nn}) + b_{\phi_k}, 0) \\
&\le \|\mathbf{a}_{\phi_k, y}\| + |\mathbf{a}_{\phi_k, \mathbf{x}_{sy}}^T \mathbf{x}_{sy} + \mathbf{a}_{\phi_k, \mathbf{g}}^T \mathbf{g}(\mathbf{x}_{nn}, \mathbf{w}_{nn}) + b_{\phi_k}|,
\end{aligned}
\tag{62}
$$

for all $j \in I_S \cup I_L$. Therefore, the negative optimal value function satisfies the conditions for Danskin's theorem Danskin (1966) (stated in Appendix C.2.1). Moreover, as there is a single unique solution to the inference problem when $\epsilon > 0$, and the quadratic objective in (16) is differentiable for all $\mathbf{w}_{sy} \in \mathbb{R}_+^r$, we can apply Corollary C.10. The optimal value function is therefore concave and differentiable with respect to the symbolic weights with

$$
\nabla_{\mathbf{w}_{sy}} V(\mathbf{w}_{sy}, \mathbf{b}(\mathbf{x}_{sy}, \mathbf{g}(\mathbf{x}_{nn}, \mathbf{w}_{nn})) = \boldsymbol{\Phi}(\mathbf{y}^*, \mathbf{x_{sy}}, \mathbf{g}(\mathbf{x_{nn}}, \mathbf{w_{nn}})).
\tag{63}
$$

Next, we show subdifferentiability of the optimal value-function with respect to the constraint constants, $\mathbf{b}(\mathbf{x}_{sy}, \mathbf{g}(\mathbf{x}_{nn}, \mathbf{w}_{nn}))$. Suppose at a setting of the neural weights $\mathbf{w}_{nn} \in \mathbb{R}^{n_g}$ there is a feasible point $\nu$ for the NeuPSL inference problem. Moreover, suppose $\nu$ strictly satisfies the $i'th$ inequality constraint of (16), i.e., $\mathbf{A}[i]\nu + \mathbf{b}(\mathbf{x}_{sy}, \mathbf{g}(\mathbf{x}_{nn}, \mathbf{w}_{nn}))[i] < 0$. Observe that the following strongly convex conic program is equivalent to the LCQP formulation of NeuPSL inference, (16):

$$
\min_{\nu \in \mathbb{R}^{n_{\mathbf{y}} + m_S + m_L}} \nu^T(\mathbf{D}(\mathbf{w}_{sy}) + \epsilon \mathbf{I})\nu + \mathbf{c}(\mathbf{w}_{sy})^T \nu + P_{\Omega \setminus i}(\nu)
\tag{64}
$$
$$
\text{s.t.} \quad \mathbf{A}[i]\nu + \mathbf{b}(\mathbf{x}_{sy}, \mathbf{g}(\mathbf{x}_{nn}, \mathbf{w}_{nn}))[i] \in \mathbb{R}_{\le 0},
$$

where $P_{\Omega \setminus i}(\nu) : \mathbb{R}^{n_{\mathbf{y}} + m_S + m_L} \to \{0, \infty\}$ is the indicator function identifying feasibility w.r.t. all the constraints of the LCQP formulation of NeuPSL inference in (16) except the $i'th$ constraint: $\mathbf{A}[i]\nu + \mathbf{b}(\mathbf{x}_{sy}, \mathbf{g}(\mathbf{x}_{nn}, \mathbf{w}_{nn}))[i] \le 0$. In other words, in the conic formulation above only the $i'th$ constraint is explicit. Note that $\mathbb{R}_{\le 0}$ is a closed convex cone in $\mathbb{R}$. Moreover, both the objective in the program and the mapping $G(\nu) := \mathbf{A}[i]\nu + \mathbf{b}(\mathbf{x}_{sy}, \mathbf{g}(\mathbf{x}_{nn}, \mathbf{w}_{nn}))[i]$ are convex. Lastly, note the

constraint qualification (43) is similar to Slater's condition in the case of (64) which is satisfied by the supposition there exists a feasible $\nu$ that strictly satisfies the $i'th$ inequality constraint of (16). Therefore, (64) satisfies the conditions of Theorem C.11. Thus, the value function is continuous in the constraint constant $\mathbf{b}(\mathbf{x}_{sy}, \mathbf{g}(\mathbf{x}_{nn}, \mathbf{w}_{nn}))[i]$ at $\mathbf{w}_{nn}$ and

$$\partial_{\mathbf{b}[i]} V(\mathbf{w}_{sy}, \mathbf{b}(\mathbf{x}_{sy}, \mathbf{g}(\mathbf{x}_{nn}, \mathbf{w}_{nn}))) = \{\mu^*[i] \,|\, \mu^* \in \operatorname*{arg\,min}_{\mu \in \mathbb{R}^{2 \cdot n_{\mathbf{y}} + m + q}_{\geq 0}} h(\mu; \mathbf{w}_{sy}, \mathbf{b}(\mathbf{x}_{sy}, \mathbf{g}(\mathbf{x}_{nn}, \mathbf{w}_{nn})))\}.$$
(65)

Moreover, when $\mathbf{b}(\mathbf{x}_{sy}, \mathbf{g}(\mathbf{x}_{nn}, \mathbf{w}_{nn}))$ is a smooth function of the neural weights $\mathbf{w}_{nn}$, then we can apply the chain rule for regular subgradients, Theorem C.8, to get

$$\hat{\partial}_{\mathbf{w}_{nn}} V(\mathbf{w}_{sy}, \mathbf{b}(\mathbf{x}_{sy}, \mathbf{g}(\mathbf{x}_{nn}, \mathbf{w}_{nn})) \supset \nabla \mathbf{b}(\mathbf{x}_{sy}, \mathbf{g}(\mathbf{x}_{nn}, \mathbf{w}_{nn})^T \partial_{\mathbf{b}} V(\mathbf{w}_{sy}, \mathbf{b}(\mathbf{x}_{sy}, \mathbf{g}(\mathbf{x}_{nn}, \mathbf{w}_{nn})).$$
(66)

To prove the optimal value function is Lipschitz smooth over $\mathbf{w}_{sy}$, it is equivalent to show it is continuously differentiable and that all gradients have bounded magnitude. To show the value function is continuously differentiable, we first apply the result asserting the minimizer is unique and a continuous function of the symbolic parameters $\mathbf{w}_{sy}$. Therefore, the optimal value function gradient is a composition of continuous functions, hence continuous in $\mathbf{w}_{sy}$. The fact that the value function has a bounded gradient magnitude follows from the fact that the decision variables $\mathbf{y}$ have a compact domain over which the gradient is finite; hence a trivial and finite upper bound exists on the gradient magnitude. $\qquad\square$

## D    Extended dual block coordinate descent

We introduce a novel block coordinate descent (BCD) algorithm for the dual LCQP formulation of NeuPSL inference in (31), a bound-constrained, strongly convex quadratic program. In this section, we omit the symbolic and neural weights from the function arguments to simplify notation. We define $U_i$, $i = 1, 2, \ldots, p$ to be a cover of the dual variable components $\{1, 2, \ldots, n_{\mathbf{y}} + m + q\}$. In practice, blocks are defined as a single dual variable corresponding to a constraint from the feasible set or a deep hinge-loss function, along with the dual variables corresponding to the bounds of the primal variables in the constraint or hinge-loss.

We will deal with a slightly more general objective,

$$h(\mu) := \frac{1}{2}\mu^T \mathbf{A}\tilde{\mathbf{D}}\mathbf{A}^T \mu + \tilde{\mathbf{c}}^T \mu,$$
(67)

from which we can recover (32) by replacing $\tilde{\mathbf{D}} \leftarrow (\mathbf{D} + \epsilon\mathbf{I})^{-1}$ and $\tilde{\mathbf{c}} \leftarrow \mathbf{A}(\mathbf{D} + \epsilon\mathbf{I})^{-1}\mathbf{c} - 2\mathbf{b}$.

We will use the superscript $\cdot^{(l)}$ to denote values in the $l$-th iteration and subscript $\cdot_{[i]}$ for the values corresponding to the block $U_i$. The row submatrix of $\mathbf{A}$ that corresponds to block $i$ is denote by $\mathbf{A}_{[i]}$.

At each iteration $l$, we choose one block $i \in \{1, 2, \ldots, p\}$ at random and compute the subvector of $\nabla h(\mu^{[l]})$ that corresponds to this block,

$$\mathbf{d}_{[i]}^{(l)} := \nabla_{[i]} h(\mu^{(l)}) = (\mathbf{A}\tilde{\mathbf{D}}\mathbf{A}^T \mu^{(l)} + \tilde{\mathbf{c}})_{[i]}.$$
(68)

Defining $\mathbf{d}^{(l)}$ to be the vector in $\mathbb{R}^N$ whose $i$th block is $\mathbf{d}_{[i]}^{(l)}$ with zeros elsewhere, we perform a line search along the negative of this direction. Note that

$$h(\mu^{(l)} - \alpha\mathbf{d}^{(l)}) = \frac{1}{2}\alpha^2 \mathbf{d}^{(l)T} \mathbf{A}\tilde{\mathbf{D}}\mathbf{A}^T \mathbf{d}^{(l)} - \alpha\mathbf{d}^{(l)T}(\mathbf{A}\tilde{\mathbf{D}}\mathbf{A}^T \mu^{(l)} + \tilde{\mathbf{c}}) + \mathbf{constant}$$
(69)

$$= \frac{1}{2}\alpha^2 \mathbf{d}_{[i]}^{(l)T} \mathbf{A}_{[i]}\tilde{\mathbf{D}}\mathbf{A}_{[i]}^T \mathbf{d}_{[i]}^{(l)} - \alpha\mathbf{d}_{[i]}^{(l)T} \mathbf{d}_{[i]}^{(l)} + \mathbf{constant}.$$
(70)

The unconstrained minimizer of this expression is

$$\alpha_l^* = \frac{\mathbf{d}_{[i]}^{(l)T} \mathbf{d}_{[i]}^{(l)}}{\mathbf{d}_{[i]}^{(l)T} \mathbf{A}_{[i]}\tilde{\mathbf{D}}\mathbf{A}_{[i]}^T \mathbf{d}_{[i]}^{(l)}}.$$
(71)

Given the nonnegativity constraints, we also need to ensure that $\mu_{[i]}^{(l)} - \alpha \mathbf{d}_{[i]}^{(l)} \geq 0$. Therefore, our choice of steplength is

$$\alpha_l = \min \left\{ \alpha_l^*, \min_{j \in U_i : \mathbf{d}_j^{(l)} > 0} \frac{\mu_j^{(l)}}{\mathbf{d}_j^{(l)}} \right\}. \tag{72}$$

To save some computation, we introduce intermediate variables $\mathbf{f}^{(l)} := \mathbf{A}^T \mathbf{d}^{(l)} = \mathbf{A}_{[i]}^T \mathbf{d}_{[i]}^{(l)}$, and $\mathbf{m}^{(l)} := \mathbf{A}^T \mu^{(l)}$. With the intermediate variables, the updates of the BCD algorithm are:

$$\mathbf{d}_{[i]}^{(l)} \leftarrow \mathbf{A}_{[i]} \tilde{\mathbf{D}} \mathbf{m}^{(l)} + \tilde{\mathbf{c}}_{[i]}, \; \mathbf{f}^{(l)} \leftarrow \mathbf{A}_{[i]}^T \mathbf{d}_{[i]}^{(l)} \tag{73}$$

$$\mathbf{m}^{(l+1)} \leftarrow \mathbf{A}^T (\mu^{(l)} - \alpha_l \mathbf{d}^{(l)}) = \mathbf{m}^{(l)} - \alpha_l \mathbf{f}^{(l)}. \tag{74}$$

With the steplength suggested by (72), descent is guaranteed at each iteration. This property is partially why the dual BCD algorithm is effective at leveraging warmstarts which is valuable for improving the runtime of learning algorithms, as is demonstrated in Section 6.2.

---

**Algorithm 3** Dual LCQP Block Coordinate Descent

---

1: Set $l = 0$ and compute an initial feasible point $\mu^{(0)}$;
2: Compute $\mathbf{m}^{(0)} = \mathbf{A}^T \mu^{(0)}$;
3: **while** Stopping Criterion Not Satisfied **do**
4:     $S_k \leftarrow \text{Permutation}([1, 2, \ldots, p])$;
5:     **for all** $i \in S_k$ (in order) **do**
6:         Compute $\mathbf{d}_{[i]}^{(l)} \leftarrow \mathbf{A}_{[i]} \tilde{\mathbf{D}} \mathbf{m}^{(l)} + \tilde{\mathbf{c}}_{[i]}$;    $\mathbf{f}^{(l)} \leftarrow \mathbf{A}_{[i]}^T \mathbf{d}_{[i]}^{(l)}$;
7:         Compute $\alpha_l \leftarrow \min \left\{ \frac{\mathbf{d}_{[i]}^{(l)T} \mathbf{d}_{[i]}^{(l)}}{\mathbf{f}^{(l)T} \tilde{\mathbf{D}} \mathbf{f}^{(l)}}, \min_{j \in U_i : \mathbf{d}_j^{(l)} > 0} \frac{\mu_j^{(l)}}{\mathbf{d}_j^{(l)}} \right\}$
8:         $\mu_{[i]}^{(l+1)} \leftarrow \mu_{[i]}^{(l)} - \alpha_l \mathbf{d}_{[i]}^{(l)}$;    $\mu_{[j]}^{(l+1)} \leftarrow \mu_{[j]}^{(l)}$ for all $j \neq i$;
9:         $\mathbf{m}^{(l+1)} \leftarrow \mathbf{m}^{(l)} - \alpha_l \mathbf{f}^{(l)}$;
10:       $l \leftarrow l + 1$;
11:     $k \leftarrow k + 1$;

---

As strong duality holds for the LCQP formulation of deep HL-MRF inference, stopping when the primal-dual gap is below a given threshold $\delta > 0$, is a principled stopping criterion. Formally, at any iteration Algorithm 3 applied to (31), we recover an estimate of the primal variable $\mathbf{v}$ from (33) and terminate when the gap between the primal and the dual objective falls below $\delta$. The stopping criterion is checked after every permutation block has been completely iterated over.

**Connected Component Parallel D-BCD**    Oftentimes, the NeuPSL dual inference objective is additively separable over partitions of the variables. In this case, the dual BCD algorithm is parallelizable over the partitions. We propose identifying the separable components via the primal objective and constraints. More formally, prior to the primal problem instantiation, a disjoint-set data structure (Cormen et al., 2009) is initialized such that every primal variable belongs to a single unique disjoint set. Then, during instantiaion, the disjoint-set data structure is maintained to preserve the property that two primal variables exist in the same set if and only if they occur together with a non-zero coefficient in a constraint or a potential. This is achieved by merging the sets of variables in every generated constraint or potential. This process is made extremely efficient with a path compression strategy implemented to optimize finding set representatives. This parallelization strategy is empirically studied in Section 6 where we refer to it as CC D-BCD.

**Lock Free Parallel D-BCD**    In general, there may only be a few connected components in the factor graph of the inference problem. In this case, D-BCD cannot fully leverage computational resources using the CC D-BCD parallelization strategy. One solution to overcome this issue and preserve the guaranteed descent property is to lock access and updates to dual variables. In other words, processes checkout locks on the dual variables to access and update its value and corresponding statistics. Unfortunately, in practice there is too much overlap in the blocks for this form of synchronization to see runtime improvements. For this reason, we additionally propose a method that sacrifices the theoretical guaranteed descent property of the dual BCD algorithm for significant

runtime improvements. Our approach is inspired by lock free parallelization strategies in optimization literature (Bertsekas & N. Tsitsiklis, 1989; Recht et al., 2011; Liu et al., 2015). Specifically, rather than having processes checkout locks on dual variables for the entire iteration, we only assume dual and intermediate variable updates are atomic. This assumption ensures the dual variables and intermediate variables are synchronized across processes. However, the steplength subproblem solution and the gradient may be incorrect. Despite this, in Section 6.1 we show this distributed variant of the dual BCD algorithm consistently finds a solution satisfying the stopping criterion and realizes significant runtime improvements over the CC D-BCD algorithm in some datasets.

## E   EXTENDED EMPIRICAL EVALUATION

In this section, we provide additional details on the datasets and NeuPSL models used in experiments, hardware used to run experiments, an additional evaluation on the effect of the LCQP regularization on the prediction performance of NeuPSL, more inference runtime experiments, and the hyperparameter details for all the experiments in the main paper.

### E.1   DATASETS AND NEUPSL MODELS

In this section, we provide additional information on all five evaluation datasets and corresponding NeuPSL models.

#### E.1.1   4FORUMS AND CREATEDEBATE

Stance-4Forums and Stance-CreateDebate are two datasets containing dialogues from online debate websites: `4forums.com` and `createdebate.com`, respectively. In this paper, we study stance classification, i.e., the task of identifying the stance of a speaker in a debate as being for or against.

The 5 data splits and the NeuPSL model we evaluate in this paper originated from Sridhar et al. (2015). The data and NeuPSL models are available at: `https://github.com/linqs/psl-examples/tree/main/stance-4forums` and `https://github.com/linqs/psl-examples/tree/main/stance-createdebate`.

#### E.1.2   EPINIONS

Epinions is a trust network with $2,000$ individuals connected by $8,675$ directed edges representing whether they know each other and whether they trust each other Richardson et al. (2003). We study link prediction, i.e., we predict if two individuals trust each other.

In each of the 5 data splits, the entire network is available, and the prediction performance is measured on $\frac{1}{8}$ of the trust labels. The remaining set of labels are available for training. We use The NeuPSL model from Bach et al. (2017). The data and NeuPSL model are available at `https://github.com/linqs/psl-examples/tree/main/epinions`.

#### E.1.3   CITESEER AND CORA

Citeseer and Cora are citation networks introduced by Sen et al. (2008). For Citeseer, $3,312$ documents are connected by $4,732$ edges representing citation links. For Cora, $2,708$ documents are connected by $5,429$ edges representing citation links. We study node classification, i.e., we classify the documents into one of 6 topics for Citeseer and 7 topics for Cora.

We study two different data settings for evaluations. For the inference and learning runtime experiments and the HL-MRF learning prediction performance experiments, Section 6.1, Section 6.2, and Section 6.3, respectively, the data is split following Bach et al. (2017). Specifically, for each of the 5 folds, $1/2$ of the nodes are sampled and specify a graph for training, and the remaining $1/2$ of the nodes define the graph for testing. $1/2$ of the node labels are observed for both the training and test graphs. For the deep HL-MRF learning prediction performance setting, Section 6.3, for each of the 10 folds, we randomly sample $5\%$ of the node labels for training $5\%$ of the node labels for validation and $1,000$ for testing.

(a) MNIST-Add1  (b) MNIST-Add2

Figure 1: Example of MNIST-Add1 and MNIST-Add2.

Moreover, we use three different NeuPSL models for this dataset. The inference and learning runtime experiment models are from Bach et al. (2017) Bach et al. (2017). The data and NeuPSL models for these experiments are available at: `https://github.com/linqs/psl-examples/tree/main/citeseer` and `https://github.com/linqs/psl-examples/tree/main/cora` for Citeseer and Cora, respectively. The models for HL-MRF learning prediction performance experiments are extended versions of those in the inference and learning runtime experiments. Specifically, a copy of each rule is made that is specialized for the topic. Moreover, topic propagation across citation links is considered for papers with differing topics. For instance, the possibility of a citation from a paper with topic $'A'$ could imply a paper is more or less likely to be topic $'B'$. The extended models are available at `https://github.com/convexbilevelnesylearning/experimentscripts/hlmrf_learning/psl-extended-examples`. The models for deep HL-MRF learning prediction performance experiments are from Pryor et al. (2023). The data and models are available at: `https://github.com/linqs/neupsl-ijcai23`.

### E.1.4 DDI

Drug-drug interaction (DDI) is a network of 315 drugs and $4,293$ interactions derived from the DrugBank database (S. Wishart et al., 2006). The edges in the drug network represent interactions and seven different similarity metrics. In this paper, we perform link prediction, i.e., we infer unknown drug-drug interactions.

The 5 data splits and the NeuPSL model we evaluate in this paper originated from Sridhar et al. (2016). The data and NeuPSL models are available at: `https://github.com/linqs/psl-examples/tree/main/drug-drug-interaction`.

### E.1.5 YELP

Yelp is a network of $34,454$ users and $3,605$ items connected by $99,049$ edges representing ratings. The task is to predict missing ratings, i.e., regression, which could be used in a recommendation system.

In each of the 5 folds, $80\%$ of the ratings are randomly sampled and available for training, and the remaining $20\%$ is held out for testing. We use The NeuPSL model from Kouki et al. (2015). The data and NeuPSL model are available at: `https://github.com/linqs/psl-examples/tree/main/yelp`.

### E.1.6 MNIST-ADDITION

MNIST Addition is a canonical NeSy image classification dataset first introduced by Manhaeve et al. (2018). In MNIST-Addition, models must determine the sum of two lists of MNIST images, for example, $\lceil 3 \rceil + \lceil 5 \rceil = 8$. The challenge stems from the lack of labels for the MNIST images; only the final sum of the equation is provided during training, $8$ in this example. 5 MNIST-Addition train splits are generated by randomly sampling, without replacement, $n \in \{600, 6,000, 50,000\}$ unique MNIST images from the original MNIST dataset and converted to MNIST additions. Specifically, additions are created by creating $n/2$ non-overlapping pairs of digits from the sample for MNIST-Add1 and $n/4$ non-overlapping sets of digits from the sample for MNIST-Add2. Then the MNIST image labels are then added together, as shown in Fig. 1, to define the addition label used in the task.

| Order | Layer | Parameter | Value |
|-------|-------|-----------|-------|
| 1 | ResNet18 (He et al., 2016) | | |
| 2 | Fully Connected | Input Shape | 128 |
| | | Output Shape | 64 |
| | | Activation | ReLU |
| 3 | Fully Connected | Input Shape | 64 |
| | | Output Shape | 10 |
| | | Activation | Gumbel Softmax (Jang et al., 2017) |

Table 7: Neural architecture used in NeuPSL MNIST-Add models.

This process is repeated to create five corresponding validation and test splits, with $1,0000$ MNIST examples being sampled per test split from the original MNIST dataset.

**MNIST-Add1** The MNIST-Add1 NeuPSL model integrates the neural component summarized in Table 7 with the symbolic model summarized in Fig. 2. The symbolic model contains the following predicates:

- **NEURAL**(**Img**, **X**) The NEURAL predicate is the class probability for each image as inferred by the neural network. Img is MNIST image identifier and X is a digit class that the image may represent.

- **DIGITSUMONESPLACE**(**W**, **X**, **Y**, **Z**) The DIGITSUMONESPLACE predicate represents whether the ones place of the sum of the digits $W$, $X$, and $Y$ is $Z$. For example, substituting $0$, $1$, $2$, and $3$ for $W$, $X$, $Y$ and $Z$, the predicate value would be $1$, but substituting $1$, $1$, $2$, and $3$ for $W$, $X$, $Y$ and $Z$ would be $0$ since $1 + 1 + 2 + 3 \neq 3$.

- **DIGITSUMTENSPLACE**(**W**, **X**, **Y**, **Z**) The DIGITSUMTENSPLACE predicate represents whether the tens place of the sum of the digits $W$, $X$, and $Y$ is $Z$. For example, substituting $0$, $1$, $2$, and $0$ for $W$, $X$, $Y$ and $Z$, the predicate value would be $1$, but substituting $0$, $1$, $9$, and $0$ for $W$, $X$, $Y$ and $Z$ would be $0$ since $1 + 1 + 2 + 3 = 10$, i.e., the tens place digit of the sum is $1$ not $0$.

- **SUMPLACE**(**Img1**, **Img2**, **place**, **Z**) The SUMPLACE predicate is the probability that the digits represented in the images identified by arguments Img1 and Img2 add up to a number with a place's place of Z.

- **CARRY**(**Img1**, **Img2**, **W**) The CARRY predicate represents is the probability that the digits represented in the images identified by arguments Img1 and Img2 add up to a number with a carry value of W. For example, images representing digits $1$ and $2$ do not have a carry. These variables are considered latent in the NeuPSL model as there are no truth labels for carries.

- **POSSIBLEDIGIT**(**X**, **Z**) The POSSIBLEDIGITS predicate represents whether a digit (X) can be included in a sum that equals a number (Z). For example, POSSIBLEDIGITS$(9, 0)$ would return $0$ as no positive digit when added to $9$ will equal $0$. Conversely, POSSIBLEDIGITS$(9, 17)$ would return $1$ as $8$ added to $9$ equals $17$.

- **IMAGESUM**(**Img1**, **Img2**, **Z**) The IMAGESUM predicate is the probability that the digits represented by the images specified by Img1 and Img2 sum up to the number indicated by the argument Z. This predicate instantiates decision variables, i.e., variables from this predicate are not fixed during inference and learning as described in the NeSy EBM, NeuPSL, and Inference and Learning sections.

- **PLACEDREPRESENTATION**($\mathbf{Z}_{10}$, $\mathbf{Z}_1$, **Z**) The PLACEDREPRESENTATION predicate represents whether the number $Z$ has tens place digit $Z_{10}$ and ones place digit $Z_1$.

**MNIST-Add2** The MNIST-Add2 NeuPSL model integrates the neural component summarized in Table 7 with the symbolic model summarized in Fig. 3. The symbolic model contains the following predicates:

- **SUMPLACE**(**Img1**, **Img2**, **Img3**, **Img4**, **place**, **Z**) The SUMPLACE predicate is the probability that the digits represented in the images identified by arguments Img1, Img2, Img3, and Img4 add up to a number with a place's place of Z.

$w_1 : \text{DIGITSUMONESPLACE}('0', \text{X}, \text{Y}, \text{Z}) \land \text{NEURAL}(\text{Img1}, \text{X}) \land \text{NEURAL}(\text{Img2}, \text{Y}) \rightarrow \text{SUMPLACE}(\text{Img1}, \text{Img2}, '1', \text{Z})$

$w_2 : \text{DIGITSUMTENSPLACE}('0', \text{X}, \text{Y}, \text{Z}) \land \text{NEURAL}(\text{Img1}, \text{X}) \land \text{NEURAL}(\text{Img2}, \text{Y}) \rightarrow \text{SUMPLACE}(\text{Img1}, \text{Img2}, '10', \text{Z})$

$w_3 : \text{DIGITSUMONESPLACE}(\text{W}, '0', '0', \text{Z}) \land \text{CARRY}(\text{Img1}, \text{Img2}, \text{W}) \rightarrow \text{SUMPLACE}(\text{Img1}, \text{Img2}, '10', \text{Z})$

$w_4 : \text{SUMPLACE}(\text{ImageId1}, \text{ImageId2}, '1', \text{Z}_1) \land \text{SUMPLACE}(\text{ImageId1}, \text{ImageId2}, '10', \text{Z}_{10})$
$\qquad \land \text{PLACEDREPRESENTATION}(\text{Z}_{10}, \text{Z}_1, \text{Z}) \rightarrow \text{IMAGESUM}(\text{ImageId1}, \text{ImageId2}, \text{Z})$

Figure 2: Summarized NeuPSL MNIST-Add1 Symbolic Model. The full model is available at: `https://github.com/convexbilevelnesylearning/experimentscripts/mnist_addition/neupsl_models`.

$w_1 : \text{DIGITSUMONESPLACE}('0', \text{X}, \text{Y}, \text{Z}) \land \text{NEURAL}(\text{Img2}, \text{X}) \land \text{NEURAL}(\text{Img4}, \text{Y}) \rightarrow \text{SUMPLACE}(\text{Img1}, \text{Img2}, \text{Img3}, \text{Img4} '1', \text{Z})$

$w_2 : \text{DIGITSUMTENSPLACE}('0', \text{X}, \text{Y}, \text{Z}) \land \text{NEURAL}(\text{Img2}, \text{X}) \land \text{NEURAL}(\text{Img4}, \text{Y}) \rightarrow \text{CARRY}(\text{Img2}, \text{Img4}, \text{Z})$

$w_3 : \text{DIGITSUMONESPLACE}(\text{W}, \text{X}, \text{Y}, \text{Z}) \land \text{NEURAL}(\text{Img1}, \text{X}) \land \text{NEURAL}(\text{Img3}, \text{Y}) \land \text{CARRY}(\text{Img2}, \text{Img4}, \text{W})$
$\qquad \rightarrow \text{SUMPLACE}(\text{Img1}, \text{Img2}, \text{Img3}, \text{Img4} '10', \text{Z})$

$w_4 : \text{DIGITSUMTENSPLACE}(\text{W}, \text{X}, \text{Y}, \text{Z}) \land \text{NEURAL}(\text{Img1}, \text{X}) \land \text{NEURAL}(\text{Img3}, \text{Y}) \land \text{CARRY}(\text{Img2}, \text{Img4}, \text{W})$
$\qquad \rightarrow \text{SUMPLACE}(\text{Img1}, \text{Img2}, \text{Img3}, \text{Img4} '100', \text{Z})$

$w_5 : \text{DIGITSUMONESPLACE}(\text{W}, '0', '0', \text{Z}) \land \text{CARRY}(\text{Img1}, \text{Img3}, \text{W}) \rightarrow \text{SUMPLACE}(\text{Img1}, \text{Img2}, \text{Img3}, \text{Img4} '100', \text{Z})$

$w_6 : \text{SUMPLACE}(\text{ImageId1}, \text{ImageId2}, \text{ImageId3}, \text{ImageId4}, '1', \text{Z}_1)$
$\qquad \land \text{SUMPLACE}(\text{ImageId1}, \text{ImageId2}, \text{ImageId3}, \text{ImageId4}, '10', \text{Z}_{10})$
$\qquad \land \text{SUMPLACE}(\text{ImageId1}, \text{ImageId2}, \text{ImageId3}, \text{ImageId4}, '100', \text{Z}_{100}) \land \text{PLACEDREPRESENTATION}(\text{Z}_{100}, \text{Z}_{10}, \text{Z}_1, \text{Z})$
$\qquad \rightarrow \text{IMAGESUM}(\text{ImageId1}, \text{ImageId2}, \text{ImageId3}, \text{ImageId4}, \text{Z})$

Figure 3: Summarized NeuPSL MNIST-Add2 Symbolic Model. The full model is available at: `https://github.com/convexbilevelnesylearning/experimentscripts/mnist_addition/neupsl_models`.

- **POSSIBLEONESDIGIT**($\mathbf{X}$, $\mathbf{Z}$) The POSSIBLEONESDIGIT predicate represents whether a digit ($\text{X}$) can be included in a sum as a ones place digit that equals a number ($\text{Z}$). For example, POSSIBLEDIGITS$(9, 0)$ would return $0$ as no positive digit when added to 9 will equal 0. Conversely, POSSIBLEDIGITS$(9, 17)$ would return $1$ as 8 added to 9 equals 17.

- **POSSIBLETENSDIGIT**($\mathbf{X}$, $\mathbf{Z}$) The POSSIBLETENSDIGITS predicate represents whether a digit ($\text{X}$) can be included in a sum as a tens place digit that equals a number ($\text{Z}$). For example, POSSIBLEDIGITS$(9, 0)$ would return $0$ as no positive digit when added to 90, 91, $\cdots$, 99 will equal 0. Conversely, POSSIBLEDIGITS$(9, 97)$ would return $1$ as 7 added to 90 equals 97, for instance.

- **IMAGESUM**($\mathbf{Img1}$, $\mathbf{Img2}$, $\mathbf{Img3}$, $\mathbf{Img4}$, $\mathbf{Z}$) The IMAGESUM predicate is the probability that the digits represented by the images specified by $\text{Img1}$, $\text{Img2}$, $\text{Img3}$, and $\text{Img4}$ sum up to the number indicated by the argument $\text{Z}$. This predicate instantiates decision variables, i.e., variables from this predicate are not fixed during inference and learning as described in the NeSy EBM, NeuPSL, and Inference and Learning sections.

- **PLACEDREPRESENTATION**($\mathbf{Z}_{100}$, $\mathbf{Z}_{10}$, $\mathbf{Z}_1$, $\mathbf{Z}$) The PLACEDREPRESENTATION predicate represents whether the number $Z$ has hundereds place digit $\text{Z}_{100}$, tens place digit $\text{Z}_{10}$ and ones place digit $\text{Z}_1$.

## E.2 HARDWARE

All timing experiments were performed on an Ubuntu 22.04.1 Linux machine with Intel Xeon Processor E5-2630 v4 at 3.10GHz and 128 GB of RAM.

### E.3  DUAL BCD AND REGULARIZATION

The regularization parameter added to the LCQP formulation of NeuPSL inference in (16) ensures strong convexity of the optimal value of the energy function. However, adding regularization makes the new formulation an approximation. In this section, the runtime and prediction performance of the D-BCD inference algorithm is evaluated at varying levels of regularization to understand its effect on NeuPSL inference. The regularization parameter varies in the range $\epsilon \in \{100, 10, 1, 0.1, 0.01\}$. The D-BCD algorithm is stopped when the primal-dual gap drops below $\delta = 0.1$ Inference time is provided in seconds, and the performance metric is consistent with Table 1. Results are provided in Table 8.

Table 8: D-BCD Inference time in seconds and prediction performance with varying values for the LCQP regularization parameter $\epsilon$.

| Dataset | $\epsilon$ | Time (sec) | Perf. |
|---|---|---|---|
| CreateDebate (AUROC) | 100 | $0.02 \pm 0.01$ | $64.77 \pm 10.61$ |
| | 10 | $0.02 \pm 0.01$ | $64.83 \pm 10.53$ |
| | 1 | $0.02 \pm 0.01$ | $64.74 \pm 10.67$ |
| | 0.1 | $0.05 \pm 0.02$ | $65.39 \pm 9.07$ |
| | 0.01 | $0.42 \pm 0.51$ | $66.01 \pm 9.35$ |
| 4Forums (AUROC) | 100 | $0.11 \pm 0.02$ | $61.31 \pm 6.17$ |
| | 10 | $0.10 \pm 0.03$ | $61.26 \pm 6.16$ |
| | 1 | $0.09 \pm 0.01$ | $61.12 \pm 6.18$ |
| | 0.1 | $0.43 \pm 0.11$ | $62.73 \pm 5.46$ |
| | 0.01 | $7.11 \pm 3.05$ | $62.31 \pm 5.47$ |
| Epinions (AUROC) | 100 | $0.33 \pm 0.05$ | $72.59 \pm 2.27$ |
| | 10 | $0.28 \pm 0.04$ | $72.69 \pm 2.21$ |
| | 1 | $0.33 \pm 0.05$ | $74.24 \pm 1.95$ |
| | 0.1 | $1.08 \pm 0.16$ | $77.05 \pm 1.06$ |
| | 0.01 | $5.21 \pm 0.37$ | $77.45 \pm 0.70$ |
| Citeseer (Accuracy) | 100 | $0.95 \pm 0.14$ | $71.28 \pm 1.31$ |
| | 10 | $1.00 \pm 0.12$ | $71.28 \pm 1.30$ |
| | 1 | $1.48 \pm 0.29$ | $71.59 \pm 1.01$ |
| | 0.1 | $7.01 \pm 1.57$ | $71.75 \pm 1.10$ |
| | 0.01 | $62.41 \pm 14.67$ | $71.92 \pm 1.09$ |
| Cora (Accuracy) | 100 | | |
| | 10 | | |
| | 1 | $7.36 \pm 4.19$ | $81.48 \pm 1.70$ |
| | 0.1 | $42.24 \pm 25.06$ | $81.88 \pm 1.82$ |
| | 0.01 | $269.45 \pm 49.50$ | $81.79 \pm 1.72$ |
| DDI (AUROC) | 100 | $24.56 \pm 0.25$ | $94.85 \pm 0.00$ |
| | 10 | $29.23 \pm 0.59$ | $94.85 \pm 0.00$ |
| | 1 | $47.15 \pm 0.95$ | $94.82 \pm 0.00$ |
| | 0.1 | $280.62 \pm 5.19$ | $94.80 \pm 0.00$ |
| | 0.01 | $266.07 \pm 42.68$ | $94.81 \pm 0.00$ |
| Yelp (MAE) | 100 | $105.60 \pm 5.03$ | $0.23 \pm 0.01$ |
| | 10 | $3,239 \pm 81$ | $0.22 \pm 0.01$ |
| | 1 | $3,227 \pm 54$ | $0.19 \pm 0.01$ |
| | 0.1 | $421 \pm 202$ | $0.18 \pm 0.00$ |
| | 0.01 | $2,472 \pm 297$ | $0.18 \pm 0.00$ |

Table 8 shows there is a consistent correlation between the LCQP regularization parameter and the runtime and performance of inference. As $\epsilon$ increases, there is a significant decrease in the runtime performance as the D-BCD algorithm can find a solution with a gradient meeting the stopping criterion in fewer iterations. Notably, for the Citeseer inference problem, the D-BCD algorithm realizes a roughly $45\times$ speedup. On the other hand, while the runtime performance improves with increasing $\epsilon$, the prediction performance can sometimes decay. There is a tradeoff between runtime and prediction performance when setting the $\epsilon$ regularization parameter.

### E.4  EXTENDED INFERENCE RUNTIME

Table 9: Inference time in seconds for each inference optimization technique.

| | Gurobi | GD | ADMM | CC D-BCD | LF D-BCD |
|---|---|---|---|---|---|
| Epinions | $0.46 \pm 0.01$ | $34.63 \pm 0.33$ | $0.36 \pm 0.041$ | $1.84 \pm 0.4$ | $\mathbf{0.26 \pm 0.04}$ |
| Citeseer | $0.66 \pm 0.08$ | $47.17 \pm 0.61$ | $0.63 \pm 0.07$ | $1.36 \pm 0.24$ | $\mathbf{0.49 \pm 0.08}$ |
| Cora | $\mathbf{0.71 \pm 0.08}$ | $48.66 \pm 1.24$ | $\mathbf{0.71 \pm 0.07}$ | $6.46 \pm 3.5$ | $0.79 \pm 0.19$ |
| Yelp | $7.38 \pm 0.20$ | $6,961 \pm 46$ | $\mathbf{6.37 \pm 1.19}$ | $48.44 \pm 3.82$ | $7.58 \pm 0.48$ |

Table 10: Hyperparameter ranges and final values for the inference runtime experiments.

| Dataset | Parameter | Range | Final Value |
|---|---|---|---|
| CreateDebate | ADMM Step Length | $\{10.0, 1.0, 0.1, 0.01\}$ | 1.0 |
|  | LCQP Regularization | $\{100, 10, 1, 0.1, 0.01\}$ | 0.1 |
| 4Forums | ADMM Step Length | $\{10.0, 1.0, 0.1, 0.01\}$ | 1.0 |
|  | LCQP Regularization | $\{100, 10, 1, 0.1, 0.01\}$ | 0.1 |
| Epinions | GD Step Length | $\{10.0, 1.0, 0.1, 0.01, 0.001\}$ | 0.01 |
|  | ADMM Step Length | $\{10.0, 1.0, 0.1, 0.01\}$ | 0.1 |
|  | LCQP Regularization | $\{100, 10, 1, 0.1, 0.01\}$ | 0.1 |
| Citeseer | GD Step Length | $\{10.0, 1.0, 0.1, 0.01, 0.001\}$ | 0.1 |
|  | ADMM Step Length | $\{10.0, 1.0, 0.1, 0.01\}$ | 10.0 |
|  | LCQP Regularization | $\{100, 10, 1, 0.1, 0.01\}$ | 10.0 |
| Cora | GD Step Length | $\{10.0, 1.0, 0.1, 0.01, 0.001\}$ | 0.1 |
|  | ADMM Step Length | $\{10.0, 1.0, 0.1, 0.01\}$ | 10.0 |
|  | LCQP Regularization | $\{100, 10, 1, 0.1, 0.01\}$ | 10.0 |
| DDI | ADMM Step Length | $\{10.0, 1.0, 0.1, 0.01\}$ | 1.0 |
|  | LCQP Regularization | $\{100, 10, 1, 0.1, 0.01\}$ | 10.0 |
| Yelp | GD Step Length | $\{10.0, 1.0, 0.1, 0.01, 0.001\}$ | 0.001 |
|  | ADMM Step Length | $\{10.0, 1.0, 0.1, 0.01\}$ | 1.0 |
|  | LCQP Regularization | $\{100, 10, 1, 0.1, 0.01\}$ | 0.1 |
| MNIST-Add1 | ADMM Step Length | $\{10.0, 1.0, 0.1, 0.01\}$ | 1.0 |
|  | LCQP Regularization | $\{100, 10, 1, 0.1, 0.01, 0.001\}$ | 0.001 |
| MNIST-Add2 | ADMM Step Length | $\{10.0, 1.0, 0.1, 0.01\}$ | 1.0 |
|  | LCQP Regularization | $\{100, 10, 1, 0.1, 0.01, 0.001\}$ | 0.001 |

This section details the hyperparameter settings and search process for the inference runtime experiments in Section 6.1. The GD, ADMM, and D-BCD algorithms are stopped when the $L_\infty$ norm of the primal variable change between iterates is less than $0.001$. For the D-BCD algorithms, the regularization parameter from Appendix E.3 resulting in the fastest runtime and yielding a prediction performance within a standard error of the best is used. The default Gurobi optimizer hyperparameters are used. Table 10 reports the range of hyperparameters searched over and the final values. Furthermore, for the MNIST-Add1 and MNIST-Add2 models, the highest performing trained neural models for each split from the performance experiments in Section 6.3 are used.

Table 9 reports the average and standard deviation of the inference runtime for Gurobi, GD, ADMM, and D-BCD algorithms on $4$ of the datasets from Table 1. As in the main paper, we see the D-BCD algorithms are competitive with ADMM, the current state of the art optimizer for NeuPSL inference. Moreover, here we see the LF D-BCD algorithm is also competitive with Gurobi for a single round of inference.

### E.5 EXTENDED LEARNING RUNTIME

This section provides details of the hyperparameter settings for the learning runtime experiments in Section 6.2. For both learning losses, a negative log regularization with coefficient $1.0e - 3$ on the symbolic weights is added to the learning loss as suggested by Pryor et al. (2023). For ADMM inference on both learning losses, the same steplength from the inference runtime experiment is used for the first 7 datasets in Table 1. Similarly, for D-BCD inference on both learning losses, the same regularization parameter from the inference runtime experiment is used for the first 7 datasets in Table 1. For the MNIST-Add experiments, we use the regularization parameter $\epsilon = 1.0e - 3$ and ADMM steplength $1.0$ as the values were found to achieve the highest final validation prediction performance.

Mirror descent is applied to learn the symbolic weights for both SP and MSE losses. The mirror descent steplength is set to a default value of $1.0e - 3$ for the first 7 datasets in Table 1. For the MNIST-Add datasets the mirror descent steplength is set to $1.0e - 14$ as in this problem we only need to learn the neural weights. The Adam steplength for the neural component of the MNIST-Add models is set to a default value of $1.0e - 3$.

Our learning framework, Algorithm 1, is used to fit the MSE learning loss. We set the initial squared penalty parameter to a default value of $2.0$ for all datasets. Moreover, for the first 7 datasets in Table 1 we set the Moreau parameter to $0.01$, the energy loss coefficient to $0.1$, and the steplength on the target variables $\mathbf{y}$ to $0.01$. For the MNIST-Add datasets we set the Moreau parameter to $1.0e - 3$, the energy loss coefficient to $10.0$, and the steplength on the target variables $\mathbf{y}$ to $1.0e - 3$.

### E.6    EXTENDED LEARNING PREDICTION PERFORMANCE

This section details the hyperparameter settings and search process for the prediction performance experiments in Section 6.3. For all learning losses, a negative log regularization with coefficient $1.0e - 3$ on the symbolic weights is added to the learning loss as suggested by Pryor et al. (2023). The remaining hyperparameter search and setting details are described separately for the HL-MRF learning and deep HL-MRF learning experiments.

**HL-MRF Learning** The LF D-BCD algorithm is used for inference in all experiments. Moreover, the D-BCD algorithm is stopped when the primal-dual gap drops below $\delta = 1.0e - 2$ CreateDebate, 4Forums, Epinions, Citeseer, Cora, and DDI while the primal-dual threshold is set to $\delta = 1.0e - 1$ to adjust to the larger scale of the dataset. For all learning losses, the learning algorithm is stopped when the training evaluation metric stops improving after 50 epochs. For the MSE and BCE losses trained with Algorithm 1, the final objective difference tolerance was set to 0.1 for the smaller CreateDebate, 4Forums, and Epinions datasets and 1 for Citeseer, Cora, DDI, and Yelp. Moreover, the initial squared penalty coefficient is set to 2 for all datasets. The remaining hyperparameters are searched over the ranges specified in Table 11. The hyperparameter value with the best performance metric on the first fold is selected.

**Deep HL-MRF Learning**

For deep HL-MRF learning in the citation network and MNIST-Add evaluations reported in Table 5 and Table 6, respectively the validation set is used to determine when to stop the learning algorithms and what weights to use for final evaluations. Specifically, after every learning step the model performance is measured on the validation data, and when 50 consecutive steps finish without improvement, the learning algorithm is stopped. For citation network datasets, the model obtaining the best validation metric averaged across all splits are used for final test evaluation. For MNIST-Add datasets, the model obtaining the best validation metric on the first split is used for final test evaluation across all splits. Table 12 and Table 13 report the range of hyperparameters searched over and the final values resulting in the highest validation prediction performance for citation network datasets and MNIST-Add datasets, respectively.

Table 11: Hyperparameter ranges and final values for the HL-MRF learning prediction performance experiments in Table 4.

| Dataset | Learning Loss | Parameter | Range | Final Value |
|---|---|---|---|---|
| **CreateDebate** | Energy | Mirror Descent Step Length | $\{1.0e-3, 1.0e-2\}$ | $1.0e-3$ |
| | | LCQP Regularization | $\{1.0e-3, 1.0e-2\}$ | $1.0e-2$ |
| | SP | Mirror Descent Step Length | $\{1.0e-3, 1.0e-2\}$ | $1.0e-2$ |
| | | LCQP Regularization | $\{1.0e-3, 1.0e-2\}$ | $1.0e-3$ |
| | MSE | Mirror Descent Step Length | $\{1.0e-3, 1.0e-2\}$ | $1.0e-2$ |
| | | y Step Length | $\{1.0e-2, 1.0e-1\}$ | $1.0e-1$ |
| | | Moreau Parameter | $\{1.0e-3, 1.0e-2, 1.0e-1\}$ | $1.0e-2$ |
| | | Energy Loss Coefficient | $\{0, 1.0e-1, 1, 10\}$ | $0.1$ |
| | | LCQP Regularization | $\{1.0e-3, 1.0e-2\}$ | $1.0e-3$ |
| | BCE | Mirror Descent Step Length | $\{1.0e-3, 1.0e-2\}$ | $1.0e-3$ |
| | | y Step Length | $\{1.0e-2, 1.0e-1\}$ | $1.0e-1$ |
| | | Moreau Parameter | $\{1.0e-3, 1.0e-2, 1.0e-1\}$ | $1.0e-2$ |
| | | Energy Loss Coefficient | $\{0, 1.0e-1, 1, 10\}$ | $10$ |
| | | LCQP Regularization | $\{1.0e-3, 1.0e-2\}$ | $1.0e-2$ |
| **4Forums** | Energy | Mirror Descent Step Length | $\{1.0e-3, 1.0e-2\}$ | $1.0e-3$ |
| | | LCQP Regularization | $\{1.0e-3, 1.0e-2\}$ | $1.0e-3$ |
| | SP | Mirror Descent Step Length | $\{1.0e-3, 1.0e-2\}$ | $1.0e-3$ |
| | | LCQP Regularization | $\{1.0e-3, 1.0e-2\}$ | $1.0e-3$ |
| | MSE | Mirror Descent Step Length | $\{1.0e-3, 1.0e-2\}$ | $1.0e-3$ |
| | | y Step Length | $\{1.0e-2, 1.0e-1\}$ | $1.0e-2$ |
| | | Moreau Parameter | $\{1.0e-3, 1.0e-2, 1.0e-1\}$ | $1.0e-3$ |
| | | Energy Loss Coefficient | $\{0, 1.0e-1, 1, 10\}$ | $0$ |
| | | LCQP Regularization | $\{1.0e-3, 1.0e-2\}$ | $1.0e-3$ |
| | BCE | Mirror Descent Step Length | $\{1.0e-3, 1.0e-2\}$ | $1.0e-3$ |
| | | y Step Length | $\{1.0e-2, 1.0e-1\}$ | $1.0e-2$ |
| | | Moreau Parameter | $\{1.0e-3, 1.0e-2, 1.0e-1\}$ | $1.0e-3$ |
| | | Energy Loss Coefficient | $\{0, 1.0e-1, 1, 10\}$ | $0$ |
| | | LCQP Regularization | $\{1.0e-3, 1.0e-2\}$ | $1.0e-3$ |
| **Epinions** | Energy | Mirror Descent Step Length | $\{1.0e-3, 1.0e-2\}$ | $1.0e-3$ |
| | | LCQP Regularization | $\{1.0e-3, 1.0e-2\}$ | $1.0e-3$ |
| | SP | Mirror Descent Step Length | $\{1.0e-3, 1.0e-2\}$ | $1.0e-3$ |
| | | LCQP Regularization | $\{1.0e-3, 1.0e-2\}$ | $1.0e-3$ |
| | MSE | Mirror Descent Step Length | $\{1.0e-3, 1.0e-2\}$ | $1.0e-2$ |
| | | y Step Length | $\{1.0e-2, 1.0e-1\}$ | $1.0e-1$ |
| | | Moreau Parameter | $\{1.0e-3, 1.0e-2, 1.0e-1\}$ | $1.0e-2$ |
| | | Energy Loss Coefficient | $\{0, 1.0e-1, 1, 10\}$ | $0.1$ |
| | | LCQP Regularization | $\{1.0e-3, 1.0e-2\}$ | $1.0e-2$ |
| | BCE | Mirror Descent Step Length | $\{1.0e-3, 1.0e-2\}$ | $1.0e-2$ |
| | | y Step Length | $\{1.0e-2, 1.0e-1\}$ | $1.0e-1$ |
| | | Moreau Parameter | $\{1.0e-3, 1.0e-2, 1.0e-1\}$ | $1.0e-2$ |
| | | Energy Loss Coefficient | $\{0, 1.0e-1, 1, 10\}$ | $1$ |
| | | LCQP Regularization | $\{1.0e-3, 1.0e-2\}$ | $1.0e-2$ |
| **Citeseer** | Energy | Mirror Descent Step Length | $\{1.0e-3, 1.0e-2\}$ | $1.0e-3$ |
| | | LCQP Regularization | $\{1.0e-3, 1.0e-2\}$ | $1.0e-2$ |
| | SP | Mirror Descent Step Length | $\{1.0e-3, 1.0e-2\}$ | $1.0e-3$ |
| | | LCQP Regularization | $\{1.0e-3, 1.0e-2\}$ | $1.0e-3$ |
| | MSE | Mirror Descent Step Length | $\{1.0e-3, 1.0e-2\}$ | $1.0e-2$ |
| | | y Step Length | $\{1.0e-2, 1.0e-1\}$ | $1.0e-2$ |
| | | Moreau Parameter | $\{1.0e-3, 1.0e-2, 1.0e-1\}$ | $1.0e-2$ |
| | | Energy Loss Coefficient | $\{0, 1.0e-1, 1, 10\}$ | $1$ |
| | | LCQP Regularization | $\{1.0e-3, 1.0e-2\}$ | $1.0e-2$ |
| | BCE | Mirror Descent Step Length | $\{1.0e-3, 1.0e-2\}$ | $1.0e-2$ |
| | | y Step Length | $\{1.0e-2, 1.0e-1\}$ | $1.0e-1$ |
| | | Moreau Parameter | $\{1.0e-3, 1.0e-2, 1.0e-1\}$ | $1.0e-3$ |
| | | Energy Loss Coefficient | $\{0, 1.0e-1, 1, 10\}$ | $0$ |
| | | LCQP Regularization | $\{1.0e-3, 1.0e-2\}$ | $1.0e-3$ |
| **Cora** | Energy | Mirror Descent Step Length | $\{1.0e-3, 1.0e-2\}$ | $1.0e-3$ |
| | | LCQP Regularization | $\{1.0e-3, 1.0e-2\}$ | $1.0e-2$ |
| | SP | Mirror Descent Step Length | $\{1.0e-3, 1.0e-2\}$ | $1.0e-3$ |
| | | LCQP Regularization | $\{1.0e-3, 1.0e-2\}$ | $1.0e-3$ |
| | MSE | Mirror Descent Step Length | $\{1.0e-3, 1.0e-2\}$ | $1.0e-2$ |
| | | y Step Length | $\{1.0e-2, 1.0e-1\}$ | $1.0e-1$ |
| | | Moreau Parameter | $\{1.0e-3, 1.0e-2, 1.0e-1\}$ | $1.0e-2$ |
| | | Energy Loss Coefficient | $\{0, 1.0e-1, 1, 10\}$ | $0.1$ |
| | | LCQP Regularization | $\{1.0e-3, 1.0e-2\}$ | $1.0e-3$ |
| | BCE | Mirror Descent Step Length | $\{1.0e-3, 1.0e-2\}$ | $1.0e-2$ |
| | | y Step Length | $\{1.0e-2, 1.0e-1\}$ | $1.0e-1$ |
| | | Moreau Parameter | $\{1.0e-3, 1.0e-2, 1.0e-1\}$ | $1.0e-2$ |
| | | Energy Loss Coefficient | $\{0, 1.0e-1, 1, 10\}$ | $0.1$ |
| | | LCQP Regularization | $\{1.0e-3, 1.0e-2\}$ | $1.0e-3$ |
| **DDI** | Energy | Mirror Descent Step Length | $\{1.0e-3, 1.0e-2\}$ | $1.0e-3$ |
| | | LCQP Regularization | $\{1.0e-3, 1.0e-2\}$ | $1.0e-2$ |
| | SP | Mirror Descent Step Length | $\{1.0e-3, 1.0e-2\}$ | $1.0e-3$ |
| | | LCQP Regularization | $\{1.0e-3, 1.0e-2\}$ | $1.0e-2$ |
| | MSE | Mirror Descent Step Length | $\{1.0e-3, 1.0e-2\}$ | $1.0e-3$ |
| | | y Step Length | $\{1.0e-2, 1.0e-1\}$ | $1.0e-1$ |
| | | Moreau Parameter | $\{1.0e-3, 1.0e-2, 1.0e-1\}$ | $1.0e-3$ |
| | | Energy Loss Coefficient | $\{0, 1.0e-1, 1, 10\}$ | $0.1$ |
| | | LCQP Regularization | $\{1.0e-3, 1.0e-2\}$ | $1.0e-2$ |
| | BCE | Mirror Descent Step Length | $\{1.0e-3, 1.0e-2\}$ | $1.0e-2$ |
| | | y Step Length | $\{1.0e-2, 1.0e-1\}$ | $1.0e-2$ |
| | | Moreau Parameter | $\{1.0e-3, 1.0e-2, 1.0e-1\}$ | $1.0e-2$ |
| | | Energy Loss Coefficient | $\{0, 1.0e-1, 1, 10\}$ | $0.1$ |
| | | LCQP Regularization | $\{1.0e-3, 1.0e-2\}$ | $1.0e-2$ |
| **Yelp** | Energy | Mirror Descent Step Length | $\{1.0e-3, 1.0e-2\}$ | $1.0e-3$ |
| | | LCQP Regularization | $\{1.0e-3, 1.0e-2\}$ | $1.0e-2$ |
| | SP | Mirror Descent Step Length | $\{1.0e-3, 1.0e-2\}$ | $1.0e-3$ |
| | | LCQP Regularization | $\{1.0e-3, 1.0e-2\}$ | $1.0e-2$ |
| | MSE | Mirror Descent Step Length | $\{1.0e-3, 1.0e-2\}$ | $1.0e-3$ |
| | | y Step Length | $\{1.0e-2, 1.0e-1\}$ | $1.0e-2$ |
| | | Moreau Parameter | $\{1.0e-3, 1.0e-2, 1.0e-1\}$ | $1.0e-1$ |
| | | Energy Loss Coefficient | $\{0, 1.0e-1, 1, 10\}$ | $10$ |
| | | LCQP Regularization | $\{1.0e-3, 1.0e-2\}$ | $1.0e-2$ |
| | BCE | Mirror Descent Step Length | $\{1.0e-3, 1.0e-2\}$ | $1.0e-3$ |
| | | y Step Length | $\{1.0e-2, 1.0e-1\}$ | $1.0e-2$ |
| | | Moreau Parameter | $\{1.0e-3, 1.0e-2, 1.0e-1\}$ | $1.0e-2$ |
| | | Energy Loss Coefficient | $\{0, 1.0e-1, 1, 10\}$ | $0.1$ |
| | | LCQP Regularization | $\{1.0e-3, 1.0e-2\}$ | $1.0e-2$ |

Table 12: Hyperparameter ranges and final values for the deep HL-MRF learning prediction performance experiments on Citeseer and Cora.

| Dataset | Loss | Parameter | Range | Final Value |
|---|---|---|---|---|
| Citeseer | Energy | Mirror Descent Step Length | $\{1.0e-3, 1.0e-2\}$ | $1.0e-3$ |
| | | LCQP Regularization | $\{1.0e-3\}$ | $1.0e-3$ |
| | SP | Mirror Descent Step Length | $\{1.0e-3, 1.0e-2\}$ | $1.0e-2$ |
| | | LCQP Regularization | $\{1.0e-3\}$ | $1.0e-3$ |
| | MSE | Mirror Descent Step Length | $\{1.0e-3, 1.0e-2\}$ | $1.0e-2$ |
| | | y Step Length | $\{1.0e-3, 1.0e-2\}$ | $1.0e-2$ |
| | | Moreau Parameter | $\{1.0e-3, 1.0e-2, 1.0e-1\}$ | $1.0e-1$ |
| | | Energy Loss Coefficient | $\{1.0e-1, 1, 10\}$ | $1.0e-1$ |
| | | LCQP Regularization | $\{1.0e-3\}$ | $1.0e-3$ |
| | BCE | Mirror Descent Step Length | $\{1.0e-3, 1.0e-2\}$ | $1.0e-2$ |
| | | y Step Length | $\{1.0e-3, 1.0e-2\}$ | $1.0e-3$ |
| | | Moreau Parameter | $\{1.0e-3, 1.0e-2, 1.0e-1\}$ | $1.0e-1$ |
| | | Energy Loss Coefficient | $\{1.0e-1, 1, 10\}$ | $1.0e-1$ |
| | | LCQP Regularization | $\{1.0e-3\}$ | $1.0e-3$ |
| Cora | Energy | Mirror Descent Step Length | $\{1.0e-3, 1.0e-2\}$ | $1.0e-2$ |
| | | LCQP Regularization | $\{1.0e-3\}$ | $1.0e-3$ |
| | SP | Mirror Descent Step Length | $\{1.0e-3, 1.0e-2\}$ | $1.0e-2$ |
| | | LCQP Regularization | $\{1.0e-3\}$ | $1.0e-3$ |
| | MSE | Mirror Descent Step Length | $\{1.0e-3, 1.0e-2\}$ | $1.0e-2$ |
| | | y Step Length | $\{1.0e-3, 1.0e-2\}$ | $1.0e-3$ |
| | | Moreau Parameter | $\{1.0e-3, 1.0e-2, 1.0e-1\}$ | $1.0e-1$ |
| | | Energy Loss Coefficient | $\{1.0e-1, 1, 10\}$ | $1$ |
| | | LCQP Regularization | $\{1.0e-3\}$ | $1.0e-3$ |
| | BCE | Mirror Descent Step Length | $\{1.0e-3, 1.0e-2\}$ | $1.0e-2$ |
| | | y Step Length | $\{1.0e-3, 1.0e-2\}$ | $1.0e-2$ |
| | | Moreau Parameter | $\{1.0e-3, 1.0e-2, 1.0e-1\}$ | $1.0e-1$ |
| | | Energy Loss Coefficient | $\{1.0e-1, 1, 10\}$ | $1.0e-1$ |
| | | LCQP Regularization | $\{1.0e-3\}$ | $1.0e-3$ |

Table 13: Hyperparameter ranges and final values for the deep HL-MRF learning prediction performance experiments on MNIST-Add datasets.

| Dataset | Loss | Parameter | Range | Final Value |
|---|---|---|---|---|
| MNIST-Add1 | Energy | Mirror Descent Step Length | $\{1.0e-14\}$ | $1.0e-14$ |
| | | Adam Step Length | $\{1.0e-4, 1.0e-3\}$ | $1.0e-3$ |
| | | LCQP Regularization | $\{1.0e-3\}$ | $1.0e-3$ |
| | BCE | Mirror Descent Step Length | $\{1.0e-14\}$ | $1.0e-14$ |
| | | Adam Step Length | $\{1.0e-4, 1.0e-3\}$ | $1.0e-3$ |
| | | y Step Length | $\{1.0e-3, 1.0e-2\}$ | $1.0e-3$ |
| | | Moreau Parameter | $\{1.0e-3, 1.0e-2, 1.0e-1\}$ | $1.0e-2$ |
| | | Energy Loss Coefficient | $\{1.0e-1, 1, 10\}$ | $10$ |
| | | LCQP Regularization | $\{1.0e-3\}$ | $1.0e-3$ |
| MNIST-Add2 | Energy | Mirror Descent Step Length | $\{1.0e-14\}$ | $1.0e-14$ |
| | | Adam Step Length | $\{1.0e-4, 1.0e-3\}$ | $1.0e-3$ |
| | | LCQP Regularization | $\{1.0e-3\}$ | $1.0e-3$ |
| | BCE | Mirror Descent Step Length | $\{1.0e-14\}$ | $1.0e-14$ |
| | | Adam Step Length | $\{1.0e-4, 1.0e-3\}$ | $1.0e-4$ |
| | | y Step Length | $\{1.0e-3, 1.0e-2\}$ | $1.0e-3$ |
| | | Moreau Parameter | $\{1.0e-3, 1.0e-2, 1.0e-1\}$ | $1.0e-3$ |
| | | Energy Loss Coefficient | $\{1.0e-1, 1, 10\}$ | $10$ |
| | | LCQP Regularization | $\{1.0e-3\}$ | $1.0e-3$ |

