# OpenReview forum: "Convex and Bilevel Optimization for Neuro-Symbolic Inference and Learning"
_ICLR.cc/2024/Conference — Submitted to ICLR 2024_

### Official Review · Reviewer_2ZRW · 2023-10-19

**Soundness:** 4 excellent
**Presentation:** 3 good
**Contribution:** 3 good
**Rating:** 6
**Confidence:** 3

**Summary:**

This work formulates neuro-symbolic learning as a bilevel optimization problem, using Moreau envelopes to smooth the NeSy energy function. This yields substantial runtime improvements over competing methods.

**Strengths:**

1. The paper is nicely structured and easy to follow.

2. The approach is well-motivated and yields substantial runtime benefits.

3. The considered problem is important to the machine learning field.

**Weaknesses:**

1. Neither variant of the propsed method (CC / LF) consistently beats the ADMM baseline across all datasets. However, it is significantly faster in most cases.

2. Only one baseline was considered. As neural-symbolic systems aren't my field I'll defer to other reviewers on whether this is adequate.

**Questions:**

1. In table 2, the LF D-BCD method performs poorly for the MNIST addition tasks. The authors note that this is due to the "high number of tightly connected components." I didn't quite understand this explanation.

2. Where does the value function introduced in (7) come from? Is it learned from data?

---

> ### Author Response · Authors · 2023-11-17
>
> Thank you for your time. We are happy to learn you found the paper to be structured, motivated, and important to the research community. We have itemized responses to your comments and questions below:
>
> - Regarding the parallelization strategies for our D-BCD inference algorithm, we would like to emphasize that connected components (CC) and lock-free (LF) D-BCD algorithms should be applied in different settings. Specifically, the CC parallelization strategy should be used when there are many independent groups of variables, i.e., many connected components of the factor graph defining the inference problem. For instance, in the MNIST-Add problem, each addition example, i.e., a set of images defining an addition, creates an independent set of variables contained in a single connected component of the inference factor graph. On the other hand, LF should be applied when the factor graph is large but connected. This is the case in the citation networks like Citeseer and Cora. Furthermore, we are happy that D-BCD often outperforms ADMM in a single round of inference, as shown in Table 2, however, we feel its real strength is shown when it is applied to learning where inference is repeated, and it can leverage warm-starts, as demonstrated in Table 3.
> - In response to the comment on the experiments' baselines, we emphasize that ADMM is the state-of-the-art inference algorithm for NeuPSL. Furthermore, we have included alternative inference baselines in the extended inference runtime experiments in E.4 of the appendix. Moreover, Dual-BCD is the first inference algorithm developed for the dual LCQP formulation of NeuPSL inference presented in this work, i.e., the first algorithm to produce optimal dual variables that can be used to produce both optimal primal variables and principled learning gradients. For baselines in performance experiments, we want to underscore the primary contribution of the paper is the learning algorithm rather than the NeSy framework of NeuPSL, i.e., we are comparing our method to the incumbent training algorithm that can only minimize value-based losses. We have included other NeSy frameworks in the four datasets of Tables 5 and 6 as reference points.
> - Q1) The LF-BCD algorithm does poorly on the MNIST-Addition problem because, in this setting, the likelihood of multiple processes simultaneously updating overlapping sets of variables is high. This is because the total size of the factor graph is relatively small and made up of tightly connected components. For this reason, the LF-BCD algorithm takes longer to converge. The CC-BCD algorithm, in this case, is effective as processes work on the many disjoint groups of variables in parallel. We expect the LF-BCD performance would improve if the total size of the factor graph increased, i.e., if there were more MNIST-Addition examples.
> - Q2) The value-function is parameterized, and the goal is to learn parameter settings from data. The energy function defining the value-function integrates neural and symbolic models. The high-level architecture (the neural layers and the symbolic structure) is designed by a human using a framework like NeuPSL.

---

> > ### Comment · Reviewer_2ZRW · 2023-11-21
> >
> > Thank you to the authors for their thorough replies. My overall impression is that the paper is well-formulated and offers concrete runtime advantages over existing methods (although as noted by other reviewers the notation is pretty heavy). Unfortunately, similarly to reviewer Zeyx neurosymbolic learning is far from my field of expertise. Therefore I can't really gauge how impactful this particular approach will be, or how likely it is to inspire extensions and subsequent work.
> >
> > I'll therefore maintain my score, although if ICLR had a "7" I would select that.

---

### Official Review · Reviewer_vhBN · 2023-10-30

**Soundness:** 3 good
**Presentation:** 2 fair
**Contribution:** 3 good
**Rating:** 6
**Confidence:** 3

**Summary:**

This paper develops a general first-order gradient-based framework for end-to-end neural and symbolic parameter learning.
The framework is formulated as a bilevel optimization problem with a constrained lower-level inference problem. The inference problem minimizes a task-specific energy function, while the upper-level problem minimizes a mixed value- and minimizer-based objective. Using the value-function approach, the bilevel problem is reformulated as an inequality-constrained optimization problem, which in turn is relaxed by allowing finite violations of the constraint. Further, to deal with potential  non-differentiability of the energy function, a Moreau-envelope-based smoothening is applied. The final algorithm iteratively solves increasingly tight relaxations of the resulting smoothed optimization problem. Each relaxation is solved using a bound-constrained augmented Lagrangian algorithm.

Next, the framework is applied to neural probabilistic soft logic, in which inference is formulated as MAP on a deep hinge-loss Markov random field. The inference problem is then reformulated as a regularized LCQP, for which several continuity properties are established. An efficiently parallelizable dual block coordinate descent algorithm is proposed for solving it.
Finally, an empirical investigation shows that the proposed set of methods consistently improves inference and training runtime, while improving the accuracy.

**Strengths:**

- The reformulation of the NeSy learning problem into its relaxed and smoothened formulation (9) is quite powerful as it allows the application of first-order optimization techniques. The employed higher-level bound-constrained augmented Lagrangian method for solving it is known but applied in a new context.
- The novel formulation of NeuPSL as an LCQP and the presented efficient and parallelizable dual block coordinate descent algorithm for solving it seems to be the key contribution. The shown continuity properties are a nice addition, the detailed proofs appear correct.
- The experiments show impressive results in terms of runtime and accuracy in multiple settings. The experimental methodology is good, with detailed information on resources and hyperparameter optimization, with open-source code provided.
- Overall, this paper makes multiple contributions that are empirically shown to significantly improve the runtime and accuracy of neuro-symbolic methods. The presented framework will probably be a starting point for various potential future applications.

**Weaknesses:**

My main critique of this paper is that the presentation is in parts not very clear, especially in Section 4. Notation is not always properly defined (e.g. the prox operator), dependencies are omitted without explicitly mentioning it (e.g. in the definition of the value function), or important parts are left out of the main text (e.g. how the the dual variables in equation 10 are updated or how the Moreau envelope of the energy function is computed). See questions for additional parts that were unclear to me.

**Questions:**

- If I understand correctly, the Algorithm 1 requires computing both the Value function and the Moreau envelope at each iteration. Both require minimization over $y$, so in the case of NeuPSL are these both computed using the BCD algorithm in section 5? I.e., is the $\epsilon$ smoothening in equation 13 the same as the $\frac{1}{\rho}$ smoothening term in equation 8? In any case, I would recommend highlighting more explicitly how exactly Section 5 links into Section 4.
- At the end of Section 5.1, the authors mention that mapping primal variables to dual variables requires calculating a pseudo-inverse of the matrix A. Is this at any point required in the proposed learning algorithm? If yes, is it described anywhere?
- Are the authors aware of previous work using Moreau envelopes in the context of neuro-symbolic or structure learning with non-differentiable settings? Such links would be a useful addition to the related work. Otherwise, if this has not been used before, the novelty of this contribution could also be highlighted more.
- One of the mentioned key contributions is that the "dual BCD algorithm for NeuPSL inference [...] naturally produces statistics necessary for learning gradients". I don't undertand this statement, how are the statistics in the BCD algorithm necessary for "learning" gradients? Do you mean they help in computing the gradients required for the NeSy learning algorithm?

Remarks:
- After equation 7: "The formulation in (7) is referred to as a value-function approach in bilevel optimization literature." Citations would be great here.
- In section 5.1, the vector $b$ seems to be an affine function in the neural predictions and symbolic inputs rather than linear.

---

> ### Author Response · Authors · 2023-11-17
>
> Thank you for your thorough review. We are excited to see such an accurate summary, and we feel the actionable suggestions you gave for improving the paper's clarity have improved our submission. We have itemized responses to your comments and questions below:
>
> - We have revisited and improved the exposition of the NeSy learning algorithm in section 4. Specifically, we have substituted the proximal operator with its definition, clarified function dependencies, and emphasized the connection between the original optimization problem of inference and the Moreau envelope in section 5. Moreover, we have added a brief description of how the dual/penalty variables are updated, and we point the reader to the appendix for more details.
> - The pseudo-inverse of the constraint matrix is unnecessary for learning as long as an inference algorithm solving the LCQP dual is used to obtain optimal dual variables. Hence, the development of the dual block coordinate descent algorithm.
> - Q1) Algorithm 1 does require computing both the value-function and the Moreau envelope of the energy function. For NeuPSL, the energy function, and thus the value-function, is regularized and its minimizer is found using the dual BCD algorithm. As previously mentioned, we have added a description of the connection between the original optimization problem of NeuPSL inference and the Moreau envelope in section 5.
> - Q2) The mapping of primal to dual variables is not necessary as long as one solves the dual LCQP formulation of inference to obtain optimal dual variables needed for learning gradients. For this reason, the pseudo inverse of the constraint matrix does not need to be computed when the dual BCD algorithm is used. We have added a discussion motivating and clarifying this point in Section 5.3.
> - Q3) We are unaware of previous work using Moreau envelopes to smooth energy functions to support gradient-based parameter learning. We have emphasized this novelty in the introduction.
> - Q4) What we meant by the statement you referred to is that the dual BCD algorithm developed for the regularized LCQP formulation produces the optimal dual variables that are necessary for computing principled sub-gradients that can be used for learning. Previous inference algorithms developed for NeuPSL have only operated on the non-smooth primal formulation in equation 17 of the appendix. We edited this statement to be more direct and clear and added the original primal inference formulation back to the main body of the paper to clarify the contribution.

---

> > ### Comment · Reviewer_vhBN · 2023-11-22
> > **Answer to rebuttal**
> >
> > Thank you for answering my questions and making corresponding changes to the manuscript.
> > Unfortunately, even after the changes, I still believe the paper is really hard to digest, which is mainly due to the very heavy notation. With the current presentation I find it still difficult for the reader to figure out how exactly the various components are pieced together. However, I believe there is still a lot to gain from the content, therefore I remain with my initial rating of 6.

---

### Official Review · Reviewer_Zeyx · 2023-10-31

**Soundness:** 3 good
**Presentation:** 3 good
**Contribution:** 3 good
**Rating:** 6
**Confidence:** 1

**Summary:**

I am not sure why I got assigned this paper, which focuses on neuro-symbolic methods and is therefore fairly far away from my area of expertise. I feel only partially qualified to review it: I don't believe I bid on this paper, but if I did, this was certainly a misclick. While I am very familiar with optimization, I don't know much about neuro-symbolic methods, and this review will reflect that, but I will still do my best. I am very open to changing my mind about this paper based on feedback from the authors and other reviewers on what to direct my attention towards.

The authors study how to fuse symbolic processing with neural networks. They start with a bilevel optimization problem for neuro-symbolic learning, reformulate it as a constrained optimization problem with certain inequality-based constraints, then propose a smoothed variant of said constraints which replaces a certain energy-function with its Moreau envelope. This problem is optimized using an augmented-Lagrangian-based algorithm. The authors apply this approach to a neural probablistic soft logic model, and study optimization-theoretic properties of the resulting objective. The authors benchmark their approach on certain models, mainly against ADMM, and show that their technique performs much better in some cases.

**Strengths:**

The proposed algorithm is shown to perform significantly on a number of benchmarks compared to ADMM. This makes sense, as the authors' method is specialized to their setting, whereas ADMM is generic.

The convex-optimization-based aspects seem reasonably well-thought-out. There are a lot of choices that one could make here, and the authors' choices seem reasonable.

The paper is reasonably well-written, though at times not easy to parse for someone outside the area.

I am very interested in what the other reviewers, who will likely know much more about this area than me, will have to say, as their reviews are likely to help direct me to what are the most important parts of the work I should direct attention to, and will update my thoughts and edit my review accordingly once the time comes.

**Weaknesses:**

Way too many acronyms. I had trouble remembering what half of them stood for once I got far enough away from their definitions. This paper goes so far in the direction of using acronyms for everything, that I strongly recommend the authors go into the other extreme and remove *all* acronyms, since doing this will make the paper easier to read.

Very little seems to actually happen in Section 2. The paper would be improved by reviewing some of the technical points in more detail, otherwise the description is so high-level that it is almost meaningless. In particular, the differences between implicit-differentiation-based methods and value-function approaches are not sufficiently explained, and are not clear to me even though I know exactly how to differentiate through an optimization problem using envelope theorems and similar. This is much more background than many readers will have, so if I'm confused, chances are most people will be. Please see questions.

The main technical sections are notation-heavy, and are quite complex to read. Theorem 5.3 in particular is somewhat hard-to-parse and takes a third of a page, in total, to state, even though its first main claim is a simply saying that a certain objective is convex-concave, and the other claims are also relatively simple.

The evaluation is purely quantitative: the authors show better numbers on tables compared to alternatives. While this is a valid way to evaluate, it also gives a much weaker idea of what is going on compared to evaluations that are not table-based and consider factors other than performance metrics. Nothing I saw in the experimental section rules out a situation of the form "none of the algorithms work, but the authors' is slightly less broken" - I would like to see some evidence that this isn't the case.
* I've previously seen misleading results like this in reinforcement learning papers where an agent achieved "good performance" through random actions that were slightly-more-aligned with the objective compared to baselines, but was so far from correct behavior that most people would reasonably view both the method and the baselines as equally bad, and the differences in metrics as meaningless. In total, how do we know the neuro-symbolic system is performing as it should in this setting?

**Questions:**

I do not understand the described difference between implicit-differentiation-based methods and value-function approaches.
* Is a value-function approach one where the lower-level objective is solved either analytically or to convergence, after which one applies a suitable envelope theorem approach to calculate the gradient of the objective using the optimal value?
* In contrast, is an implicit-differentiation approach one where we do not solve the inner optimization problem to convergence? Or am I completely confused by what you mean by this distinction here?

Why is the constraint in (7) an inequality constraint, rather than an equality constraint? The paragraph directly after it mentions an equality constraint. Is this a typo?

Is there a way to evaluate the performance of the resulting neuro-symbolic system qualitatively, to ensure it behaves in the manner that the algorithmic designer expects it does? Is there some kind of sanity check one could do to guard against the "all the methods fail at the given task, but ours fails with slightly better numbers" potential failure mode? While I certainly have no direct evidence that this is happening, it would make me feel much better about the paper if this could be definitively ruled out via the experiments.

---

> ### Author Response · Authors · 2023-11-17
>
> Thank you for your thoughtful review. We appreciate the effort you took to carefully read and grasp the paper. We want to acknowledge the fact that although you claim this research is far from your expertise, your summary is accurate and your reviews are valuable and show you understood our motivations and methods. We have responded to each of your comments and questions below:
>
> - We have added to the structure and discussion in our related work (Section 2). Specifically, we have expanded on the definition and comparison between implicit differentiation and value-function approaches to bilevel optimization.
> - We have consolidated and simplified the statements of theorems 5.1 and 5.2.
> - The prediction performance results presented in our empirical evaluation, Section 6, improve upon the performance of NeuPSL models presented in published research. Please see Appendix E.1 for details on the NeuPSL models. Moreover, we have included prominent alternative NeSy baselines for the Citeseer, Cora, and MNIST experiments in Tables 5 and 6. This gives a relative point for comparison of the performance of NeuPSL.
> - Q1) An implicit differentiation approach derives an analytic expression for the gradient of the upper-level objective using implicit differentiation. The resulting gradient requires second-order derivatives of the lower-level objective function and constraints (if they are supported) at the lower-level minimizer. On the other hand, value-function approaches reformulate the bilevel problem as a single-level constrained program to derive gradient-based methods that only require a gradient of the optimal value of the lower-level objective function.
> - Q2) The constraint in 7 can correctly be either an inequality or equality. We intended it to be an inequality as that is becoming a standard in value-function formulations. We updated the text to match the mathematical formulation.
> - Q3) We would be happy to add additional metrics to our evaluation that you feel would better represent the performance than the ones mentioned in Table 1. However, we are unsure what types of evaluation you are referring to. We included additional NeSy methods as reference points for the four datasets in Tables 5 and 6, and we want to reiterate that the results in Table 4 improve upon the performance of NeuPSL models presented in published research.

---

> > ### Comment · Reviewer_Zeyx · 2023-11-22
> >
> > Thanks very much for your reply. From looking at the other reviews, it seems that those with lower scores have substantially more expertise in neuro-symbolic learning. Since this work is way out of my area, I will support accepting this paper only if your rebuttal and updated draft convinces those reviewers that this paper is above the bar for this subfield. I will do what I can to encourage the other reviewers to participate during the discussion phase.

---

### Official Review · Reviewer_oqJp · 2023-11-01

**Soundness:** 2 fair
**Presentation:** 1 poor
**Contribution:** 1 poor
**Rating:** 3
**Confidence:** 3

**Summary:**

The authors give an equivalent formulation of NeSy EBMs learning as a bilevel problem. Such formulation allows smooth first-order optimization.

**Strengths:**

The writing of the paper makes it very hard to identify many strengths. It is my understanding, however, that by operating at the level of NeSy EBMs, any advancement applies to (potentially) a broad class of neuro-symbolic methods.

**Weaknesses:**

- The paper is a neuro-symbolic AI approach for learning and inference with barely any mention of previous related work in the field [1, 2, 3, 4, 5, 6, ..].

- The paper is **very** hard to read with not a single figure or running example to help with the exposition.

- Empirical evaluation is only carried out on toy datasets, using very basic tasks (e.g. MNIST-addition is evaluated only using $2$ digits, which can be easily solved using existing baselines), and the numbers are presented with no extra commentary/analysis.

References:

[1] Semantic Probabilistic Layers for Neuro-Symbolic Learning. Kareem Ahmed, Stefano Teso, Kai-Wei Chang, Guy Van den Broeck, Antonio Vergari. NeurIPS 2022.

[2] A Semantic Loss Function for Deep Learning with Symbolic Knowledge. Jingyi Xu, Zilu Zhang, Tal Friedman, Yitao Liang, Guy Van den Broeck. ICML 2018.

[3] Semantic Strengthening of Neuro-Symbolic Learning. Kareem Ahmed, Kai-Wei Chang, Guy Van den Broeck. AISTATS 2023.

[4] Neuro-Symbolic Entropy Regularization. Kareem Ahmed, Eric Wang, Kai-Wei Chang, Guy Van den Broeck. UAI 2022.

[5] Coherent Hierarchical Multi−Label Classification Networks. Eleonora Giunchiglia and Thomas Lukasiewicz. NeurIPS 2022.

[6] Deep Learning with Logical Constraints. Eleonora Giunchiglia‚ Mihaela Catalina Stoian and Thomas Lukasiewicz. IJCAI 2022.

**Questions:**

- I'm struggling to understand what the problem being solved here is exactly. The second paragraph in the introduction mentions that "the predictions are not guaranteed to have an analytical form or be differentiable, and traditional deep learning techniques are not directly applicable", could you please say more about that? All the NeSy AI techniques that I am aware of: semantic loss, deepproblog, NeSy entropy, semantic strengthening, semantic probabilistic layers, neupsl, etc... are able to train the models end-to-end using back-propagation.'

- As a follow-up question: why should we be interested in developing this equivalent formulation as a bilevel problem? What could one possibly gain that improves upon the exact inference proposed in semantic loss, deepproblog, NeSy entropy and  semantic probabilistic layers?

- I am struggling to understand the results obtained on MNIST-addition. How can it be that NeuPSL, which is approximate, can outperform DeepProbLog which performs exact inference?

---

> ### Author Response · Authors · 2023-11-17
>
> Thank you for your review. Below is an itemized response to comments and questions:
>
> - We have expanded our related work section, and included in this change is the addition of a subset of the NeSy papers you have cited. However, many of these new citations belong to a subgroup of NeSy systems that enforces constraints on the structure of the symbolic model to ensure the final prediction has an explicit expression for its gradient with respect to the model parameters. In this work, we focus on the general setting where inference is an expressive and complex optimization problem, i.e., an explicit form of the gradient of the prediction, i.e., the problem solution, with respect to the model parameters does not exist or is impractical to compute. Going into detail on a tangential research direction under the large umbrella of NeSy is distracting for the reader and we maintained the focus on systems in the more general category and on bilevel optimization.
> - Regarding the comment on “toy datasets,” the MNIST-Addition dataset has become a canonical task in the NeSy community. We and other researchers have found it useful for developing and evaluating our methods. As you suggested, using entities other than digits is undoubtedly a valid possible experimental setting, and we encourage its exploration. However, we feel the seven real-world datasets, together with the two MNIST Addition settings, form an extensive empirical evaluation.
> - Q1) You are absolutely correct that a line of NeSy approaches making predictions with an explicit gradient with respect to the neural and symbolic parameters exists. This research has undoubtedly made significant strides in creating efficient and end-to-end differentiable NeSy models. However, an emerging and, in our opinion, more advanced line of NeSy research exists where predictions are solutions to complex optimization problems [1,2,3,4,5,6,7,8]. For these models, predictions are not guaranteed to have an analytic map from the inputs and parameters or even be differentiable, and traditional deep-learning techniques are not directly applicable. For instance, you cited NeuPSL as an end-to-end trainable model via backpropagation. While it is true the gradient of the neural parameters is computed using the backpropagation algorithm, the gradient propagated in the original NeuPSL publication [1] is not from a supervised loss and is not with respect to the final NeuPSL prediction. Rather the gradient is for a value-based loss. Moreover, our paper is the first to formally show the value-function of NeuPSL is sub-differentiable (Theorem 5.3) with respect to the neural network predictions. Previous research, including  [1], assumed this was the case without formal proof.
> - Q2) NeSy-EBMs are highly expressive and are used to perform large-scale joint reasoning with loopy dependencies and to ensure the satisfaction of constraints during both learning and inference. Moreover, NeSy-EBMs often do perform exact inference, contrary to your implication, and exploring this direction of research is valuable to the community.
> - Q3) Variations in performance between models in the MNIST-Addition tasks are a result of several differences. In the MNIST problem, NeuPSL inference is solved to an exact solution during both learning and final evaluation. Furthermore, the BCE loss used to train NeuPSL includes both a minimizer and a value-based loss, which can lead to data efficiency and good generalization. Lastly, we used the neural model architectures provided in the original publications for LTNs and DeepProblog, and naturally, there are variations in the implementations.
>
> References:
> 1. Connor Pryor, Charles Dickens, Eriq Augustine, Alon Albalak, William Yang Wang, and Lise Getoor. Neupsl: Neural probabilistic soft logic. In IJCAI, 2023.
> 2. Sridhar Dasarth, Sai Akhil Puranam, Karmvir Aingh Phogat, Sunil Reddy Tiyyagura, and Nigel Duffy. Deeppsl: End-to-end perception and reasoning. In IJCAI, 2023.
> 3. Giuseppe Marra, Michelangelo Diligenti, Francesco Gianini, Marco Gori, and Marco Maggini. Relational Neural Machines. In ECAI, 2020.
> 4. Samy Badreddine, Artur d’Avila Garcez, Luciano Serafini, and Michael Spranger. Logic tensor networks. AI, 303(4):103649, 2022.
> 5. Po-Wei Wang, Priya Donti, Bryan Wilder, and Zico Kolter. SATNet: Bridging Deep Learning and Logical Reasoning Using a Differentiable Satisfiability Solver. In ICML, 2019.
> 6. Brandom Amos and J. Zico Kolter. OptNet: Differentiable Optimization as a Layer in Neural Networks. In ICML, 2017.
> 7. Paolo Dragone, Stefano Teso, and Andrea Passerini. Neuro-Symbolic Constraint Programming for Structured Prediction. In NeSy, 2021.
> 8. Cristina Cornelio, Jan Stuehmer, Shell Xu Hu, and Timothy Hospedales. Learning Where and When to Reason In Neuro-Symbolic Inference. In ICLR, 2023.

---

> ### Comment · Reviewer_oqJp · 2023-11-23
>
> Thanks your response.
>
> - I'm sorry, but i'm not really sure I'm convinced by your reasoning for omit this line of work from your related works, especially in light of using the MNIST-Addition dataset and comparing against DeepProbLog, DeepStochLog and NeuPSL which belong to that very same line of work.
>
> - I'm not sure what the authors mean by "enforces constraints on the structure of the symbolic model to ensure the final prediction has an explicit expression for its gradient with respect to the model parameters", but again all these works enforce logical constraints on the *output* of the neural network.
>
> - I don't oppose the use of MNIST-Addition, rather the specific variant being used where the sum of only two digits are considered. In all the neuro-symbolic approaches that i'm aware of, the innovation comes in the probabilistic inference occurring on top of the neural networks classifier. We are already able to perform exact inference in the task of MNIST-Addition using 2 digits, which is why most recent approach have ramped up the difficulty of the task by considering more than 2 digits as it is in that regime that the exact inference approaches such as deepproblog struggle.
>
> - "more advanced line of NeSy research exists" I'm not sure I understand, or agree with such a sentiment; the SAT solver used in SATNET and OPTNet is pretty much the same machinery that powers deepproblog, semantic loss as well as many of the other publications I have cited. NeuPSL and DeepPSL and Logic tensor networks are all relaxations of semantic loss (cited above) using fuzzy logic.
>
> - "For instance, you cited NeuPSL as an end-to-end trainable model via backpropagation. While it is true the gradient of the neural parameters is computed using the backpropagation algorithm, the gradient propagated in the original NeuPSL publication [1] is not from a supervised loss and is not with respect to the final NeuPSL prediction" I agree that the gradient does not derive from labeled data, rather from minimizing the (relaxed) probability of a logical constraint (using fuzzylogic) w.r.t the network's output distribution.
>
> - Q2) I don't doubt that EBMs are highly expressive, it's not clear to me how one can reason tractably and exactly under such models, let alone condition the distribution on constraints?
>
> - Q3) I'm still not clear on how, under the same setting, NeuPSL which minimizes a relaxation of the original problem (Section B in the appendix of the Neupsl paper states: "NeuPSL operates on real-valued logic, which improves scalability but is a relaxation of the original problem"), can outperform an exact minimizer i.e. deepproblog

---

### Author Response · Authors · 2023-11-17

We thank the reviewers for their valuable time and insightful suggestions. We are excited that reviewers found our NeSy learning framework applicable to a broad class of methods and a starting point for various future applications (Reviewers oqJp, vhBN). Moreover, we are honored reviewers found our framework and optimization techniques to be “well-thought-out,” “novel,” “powerful,” and “well-motivated” (Reviewers Zeyx, vhBN, 2ZRW). Lastly, we thank the reviewers for finding areas where clarifying details (and fixing typos) could be added to strengthen the paper further. We have made significant edits to our submission to address these comments and questions.

---

### Meta-Review · Area_Chair_vQ5J · 2023-12-07

**Metareview:**

This paper considers neuro-symbolic (NeSy) parameter learning. The proposed approach for solving this problem incorporates convex optimization and bilevel optimization techniques. In particular, the authors model this problem as a bilevel optimization problem (section 4), in which an energy function called EBM (section 3) is used in both upper-level and lower-level problems. To demonstrate the capability of the proposed method, the authors specifically worked on the NeuPSL problem. Overall, we found that this paper is not well written. The presentation should be significantly improved. The motivation and the organization are not clear from the current presentation. For example, as some reviewers pointed out, it is not clear which problem is the main focus. The authors are encouraged to improve the presentation and submit to a future venue.

**Justification For Why Not Higher Score:**

The presentation is not clear.

**Justification For Why Not Lower Score:**

NA

---

### Decision · Program_Chairs · 2024-01-16

Reject